# Enhancement of hippocampal interneuron excitability by NMDA receptor positive allosteric modulation

Hao Xing[1] , Tue G. Banke[1], Lu Zhang[1], Kuai Yu[1], Chad R. Camp[2] , Russell G. Fritzemeier[3], Nicholas S. Akins[3], Srinu Paladugu[3] , Paul J. Arcoria[3], Brian R. Brady[2] , Olga Prikhodko[2], Matthew J. Kennedy[2], Dennis C. Liotta[3], Hongjie Yuan[1] and Stephen F. Traynelis[1,4]

[1]*Department of Pharmacology and Chemical Biology, Emory University School of Medicine, Atlanta, GA, USA*
[2]*Department of Pharmacology, University of Colorado Anschutz, Aurora, CO, USA*
[3]*Department of Chemistry, Emory University, Atlanta, GA, USA*
[4]*Center for Neurodegenerative Disease, Emory University, Atlanta, GA, USA*

Handling Editors: Jing-Ning Zhu & Ming Yi

The peer review history is available in the Supporting Information section of this article (https://doi.org/10.1113/JP289774#support-information-section).

The Journal of Physiology

**Abstract figure legend** EU1622-240 is a positive allosteric modulator of NMDA receptors. Administration of EU1622-240 increases both EPSPs and IPSPs in CA1 pyramidal cells. The increased EPSP was attributable to direct potentiation of GluN2B expressed by pyramidal cells. Increased inhibitory tone on pyramidal cells comes from increased

Dr. **Hao Xing** born in a small town in middle east of China, focused on the mechanism underlying pulmonary artery hypertension and got his Master of Art degree from Harbin Medical University, China, 2015. He, then trained as an electrophysiologist by Dr. Chun Jiang, obtained his Ph. D. degree from Georgia State University in 2021 with neurobiological & behavioral studies investigating respiration dysfunction in Rett Syndrome. He joined Dr. Stephen F. Traynelis's and Dr. Hongjie Yuan's lab at Emory University in 2021 to start his post-doctoral fellowship work to test the effects of subunit-targeting allosteric modulation of NMDARs in reversing epilepsy and schizophrenia.

interneuron spontaneous firing activity via potentiation of GluN2D-containing NMDA receptors, which are expressed in interneurons.

**Abstract**  *N*-Methyl-ᴅ-aspartate receptors (NMDARs) are known for their role in mediating a calcium-permeable, slow component of excitatory synaptic transmission. These receptors play important roles in multiple facets of brain functions, and their dysfunction has been implicated in neurological disease aetiology. Here, we describe the actions of a positive allosteric modulator (PAM), EU1622-240, on NMDARs within the hippocampal circuit. EU1622-240 is a pan-PAM that enhances the function of all GluN2 subunit-containing NMDARs with submicromolar potency, with the strongest effects on GluN2C- and GluN2D-containing NMDARs. Previously, we have shown that EU1622-240 enhances the maximal response, prolongs the response time course, enhances agonist potency and reduces single channel conductance. Using whole-cell patch-clamp recordings, we evaluated the effects of this PAM on both CA1 pyramidal cells and CA1 stratum radiatum interneurons in immature hippocampus. Although we observed potentiation of evoked NMDAR-mediated EPSCs on both CA1 pyramidal cells and interneurons, the PAM preferentially enhanced interneuron excitability owing to the expression of GluN2D in interneurons and increased the ratio of inhibition to excitation. This appears to result from cellular depolarization, increased spike firing and enhanced NMDAR-mediated current charge transfer in interneurons. In contrast, EU1622-240 did not detectably depolarize CA1 pyramidal cells in slices but did have modest effects when bicuculline was used to block GABAergic signalling. We also observed EU1622-240 enhancement of AMPA receptor synaptic signalling in a manner reminiscent of long-term potentiation. These data support the idea that EU1622-240 enhances interneuron function, with modest effects on the CA1 pyramidal cells, providing therapeutically beneficial effects in situations where interneuron output is diminished.

(Received 25 July 2025; accepted after revision 15 September 2025; first published online 6 October 2025)

**Corresponding author** Stephen F. Traynelis: Rollins Research Centre, 1510 Clifton Road, Atlanta, GA 30322, USA. Email: strayne@emory.edu

## Key points

- EU1622-240 is a potent positive allosteric modulator of all GluN2-containing NMDA receptors.
- EU1622-240 is active at native receptors in acute brain slices, increasing NMDA receptor-mediated charge transfer onto both CA1 principal cells and interneurons.
- Despite its actions on principal cells, EU1622-240 appears to drive preferential enhancement of interneuron function within the hippocampal network.
- EU1622-240 is also capable of increasing calcium flow into cultured hippocampal neurons, in addition to influencing AMPA receptor-mediated EPSPs that occlude conventional theta-burst-driven long-term potentiation.

## Introduction

*N*-Methyl-ᴅ-aspartate receptors (NMDARs) mediate a slow $Ca^{2+}$-permeable component of excitatory synaptic transmission throughout the CNS. NMDARs are blocked in a voltage-dependent manner by extracellular $Mg^{2+}$, hence current flow is dependent on coincident glutamate release and postsynaptic neuronal depolarization. This dependence on coincident stimuli (synaptic glutamate release plus neuronal depolarization) underlies NMDAR involvement in a host of normal brain functions, such as learning, memory and development (Hansen, Wollmuth et al. 2021). In addition, a wide range of neurological disorders have been proposed to involve aberrant NMDAR activity or, potentially, to be treated by altering NMDAR activity (Hansen, Wollmuth et al. 2021; Hanson, Yuan et al. 2024). A number of these disorders, such as schizophrenia, are hypothesized to involve reduced NMDAR function (Gao, Yang et al. 2022; Lu, Mu et al. 2024). Furthermore, given the role of NMDARs in learning and memory, disorders associated with cognitive dysfunction and memory impairment, such as

Alzheimer's disease, might be amenable to therapeutic treatment by agents that increase NMDAR function. One would expect this to improve memory formation and, therefore, quality of life. Given the potential utility of enhanced NMDAR function, a number of positive allosteric modulators (PAMs) have been described in recent years (Beckley, Aman et al. 2024; Epplin, Mohan et al. 2020; Khatri, Burger et al. 2014; Perszyk, Swanger et al. 2020; Strong, Epplin et al. 2021; Tang, Beckley et al. 2023; Wang, Brown et al. 2017; Yi, Rouzbeh et al. 2020).

The NMDAR is a heterotetrametric complex of two obligate GluN1 subunits and two GluN2 subunits or two GluN1 and two GluN3 subunits, leading to different NMDAR compositions and properties. There are four different GluN2 subunits (GluN2A, GluN2B, GluN2C and GluN2D) and two different GluN3 subunits (GluN3A and GluN3B), which are encoded by different genes (*GRIN2A*, *GRIN2B*, *GRIN2C*, *GRIN2D*, *GRIN3A* and *GRIN3B*) (Hansen, Wollmuth et al. 2021). These subunits confer distinct functional properties to the NMDAR, affecting both its $Ca^{2+}$ permeability and its response time course (Hansen, Wollmuth et al. 2021). Subunit expression is also cell-type specific, with excitatory pyramidal cells expressing GluN1, GluN2A and GluN2B (Hansen, Ogden et al. 2014) and GABAergic interneurons in cortical and hippocampal tissue expressing GluN1, GluN2A, GluN2B and GluN2D (Hanson, Yuan et al. 2024; Perszyk, DiRaddo et al. 2016; von Engelhardt, Bocklisch et al. 2015). In the mammalian hippocampus, NMDARs play important roles in both excitatory pyramidal cells and GABAergic inter-neurons. In CA1 pyramidal cells, postsynaptic NMDARs bind to synaptically released glutamate, which allows the influx of $Na^+$ and $Ca^{2+}$ into the cell. NMDAR activation can trigger long-term potentiation (LTP) of synaptic function, a process associated with synaptic plasticity and memory formation (Luscher & Malenka 2012). The activation of NMDARs on GABAergic interneurons can result in the release of GABA, which leads to the inhibition of CA1 pyramidal cells, modulation of their firing rates and control of overall network activity (Maccaferri & Dingledine 2002; Pelkey, Chittajallu et al. 2017). Thus, CA1 interneurons control the timing and synchronization of pyramidal cell firing, impact memory formation and retrieval, contribute to information processing within the hippocampus and serve as a brake on pyramidal cell firing to prevent excessive neural activity (Pelkey, Chittajallu et al. 2017; Vormstein-Schneider, Lin et al. 2020; Wester & McBain 2014).

We have recently described a new series of PAMs that can enhance the response to maximally effective concentrations of glutamate, prolong the time course of the NMDAR response and reduce the relative $Ca^{2+}$:$Na^+$ permeability ratio (Chou, Epstein et al. 2024; Fritzemeier, Akins et al. 2025; Perszyk, Swanger et al. 2020; Ullman, Perszyk et al. 2024). This series of PAMs contains a unique combination of properties, which might diminish the risk of excess excitation by virtue of the reduction in $Ca^{2+}$ permeation. In this study, we explore the actions of this series on hippocampal neuronal function and plasticity. Our results show that these compounds enhance NMDAR responses in both pyramidal cells and interneurons, with preferential actions on the excitability of interneurons. We also show that this compound can enhance calcium signalling in cultured pyramidal neurons, in addition to showing a form of prolonged enhancement of excitatory synaptic signalling that appears similar in manifestation to stimulus-induced LTP. These data highlight the utility of NMDAR potentiation in altering network function and therapeutically addressing neuropsychiatric and neuro-logical disease states.

## Methods

### Ethical approval

All animal procedures were approved by the Emory University Institutional Animal Care and Use Committee (protocol: 201 700 124), fully certified by the Association for Assessment and Accreditation of Laboratory Animal Care International, and were performed in accordance with state and federal Animal Welfare Acts and the policies of the Public Health Service. C57BL/6J mice were purchased from Jackson Laboratory (catalogue no. 000664) and bred in animal facilities at Emory University. *Grin2d* knockout (GluN2D-KO) mice were kindly provided by Dr S. Nakanishi at Kyoto University. GluN2D-KO mouse genotyping was performed by TransnetYX using the following primers and reporters: GluN2D-KO forward primer, GCCTTCTTGACGAGTTCTTCTGA; GluN2D-KO reverse primer, CACGAGGAGCATGTAGAAGACA; *Grin2d* wild-type (WT) forward primer, ATGTTGTCGATGATCCCC; and *Grin2d* WT reverse primer, CAACGACAAAATCGAGGTGATGAG. Mice were housed in ventilated cages with a 12 h–12 h dark–light cycle, at controlled temperature (22–23°C) with humidity at ∼60% and *ad libitum* access to food and water. For experimentation, animals were randomly selected from various breeder pairs or trios. GluN2D-KO mice were kept separated from heavy foot traffic and on a 10 h–14 h (light–dark) cycle to improve poor breeding.

For live cell $Ca^{2+}$-imaging experiments, animal use was conducted in accordance with guidelines approved by the Administrative Panel on Laboratory Animal Care at University of Colorado, Anschutz School of Medicine, accredited by the Association for Assessment and Accreditation of Laboratory Animal Care International (00235) and approved under the animal protocol #0300. Timed pregnant Sprague–Dawley rat dams (typically, day 16) were obtained from Charles River Laboratories

and housed in standard conditions (12 h–12 h light–dark cycle, with food and water *ad libitum*) until the litter was born.

## Chemicals

(+)-MK-801 maleate, (−)-bicuculline methiodide and NBQX were obtained from Tocris; D,L-AP5 sodium salt was obtained from HelloBio (HB0252-50mg); biocytin was obtained from Cayman; anti-streptavidin Alexa Fluor™ 546 conjugate was obtained from Thermo-Fisher; and EU1622-240 was synthesized according to published protocols (Chou, Epstein et al. 2024). All other chemicals were from Sigma. EU1622-240 was dissolved in DMSO to a stock concentration of 10 mM, and working concentrations were made at 0.1, 0.3, 1, 3, and 6 μM. Vehicle treatments of DMSO were made to match the DMSO percentage in each final working solution. Drug stocks for experiments were made fresh on the first day that testing was performed. No particulate matter was observed in the solution, and there was no further evidence of drug precipitation during the electrophysiological experiments. The stocks were stored at −20°C.

## Mouse brain hippocampal slice preparation

Coronal brain slices (280–300 μm thick) were made from WT or GluN2D-KO mice, postnatal day 17–22, of both sexes. Mice were killed by isoflurane overdose followed by decapitation and removal of the brain. The brain was rapidly placed into ice-cold artificial cerebrospinal fluid (aCSF) composed of (mM): 230 sucrose, 24 NaHCO$_3$, 10 glucose, 2.5 KCl, 1.25 NaH$_2$PO$_4$, 3 sodium pyruvate, 5 sodium L-ascorbate, 12 *N*-acetyl-cysteine, 10 MgCl$_2$ and 0.25 CaCl$_2$ saturated with 95% O$_2$–5% CO$_2$. The brain was then hemisected, glued to the removable stage and sectioned on a vibratome into 280- to 300-μm-thick slices (Leica VT1200S, Wetzlar, Germany). After sectioning, slices were transferred to a holding chamber with the same solution, except that Mg$^{2+}$ was lowered to 1.5 mM. Slices were incubated at 32°C for 30 min in this solution before returning them to room temperature for ≥1 h recovery prior to use.

## Whole-cell voltage- and current-clamp recordings

Slices were transferred to a submersion chamber, and neurons were identified via infrared light differential interference contrast (IR-DIC) on an upright microscope (BX50WI; Olympus). Slices were continuously perfused at a rate of 4 ml/min with oxygenated standard recording aCSF containing (mM): 126 NaCl, 26 NaHCO$_3$, 10 glucose, 2.5 KCl, 1.25 NaH$_2$PO$_4$, 1.5 MgSO$_4$ and 1.5 CaCl$_2$

bubbled with 95% O$_2$–5% CO$_2$. The temperature of the external solution was heated to 30–32°C by a TC-344C inline heater system (Warner Instruments). The recording chamber was flushed with ∼10 ml of 70% ethanol and ∼20 ml of diH$_2$O after each slice was recorded. Recording electrodes were made from borosilicate glass (1.50 mm outer diameter, 1.12 mm inner diameter; World Precision Instruments) via a micropipette puller (P-1000; Sutter Instruments); pipette resistances were 6–8 MΩ. Recordings were made using a Multiclamp 700B amplifier (Molecular Devices), filtered at 2 kHz using an eight-pole Bessel filter (−3 dB), and digitized at 20 kHz (Digidata 1550B) using Axon pClamp10 software.

For intrinsic and action potential firing properties, the intracellular solution contained (mM): 115 potassium gluconate, 0.6 EGTA, 2 MgCl$_2$, 2 Na$_2$ATP, 0.3 Na$_2$GTP, 10 HEPES, 5 sodium phosphocreatine, 8 KCl and 0.5% biocytin. After obtaining a whole-cell configuration, cells were dialysed for 5 min in current-clamp mode at $I = 0$ (without current injected). The liquid junction potential was not corrected for resting membrane potential (RMP), and all current-clamp responses were automatically bridge balanced using the Multiclamp 700B clamp commander software. Input resistance was calculated by using the steady-state voltage change in response to a −30 pA, 200 ms hyperpolarizing current injection period (Oginsky, Cui et al. 2014; Zhong, Cui et al. 2015) or using the slope of voltage deflections in response to 500 ms current injections from −120 to +12 pA, with $\Delta$12 pA step change at an inter-sweep interval of 2 s. The evoked action potential firing frequency was calculated in response to a 500 ms current injection every 2 s, starting at −40 pA and increasing by 20 pA until the cell displayed depolarization-induced block of firing.

For voltage-clamp recordings of evoked NMDAR-mediated excitatory postsynaptic currents (EPSCs), the electrodes were filled with internal solution containing (mM): 105 caesium gluconate, 5 CsCl, 8 NaCl, 5 sodium phosphocreatine, 5 MgCl$_2$, 2 Na-ATP, 0.3 Na-GTP, 0.6 EGTA, 5 BAPTA, 40 HEPES and 5 QX314 (pH 7.3, adjusted with CsOH, ∼290 mOsmol/kg). Both 10 μM gabazine and 10 μM NBQX were added to the external solution to block GABA$_A$ receptors and AMPA receptors, respectively. Cells were held at +40 mV, and NMDAR-mediated EPSCs were recorded by injecting 50–120 μA of current for 0.1 ms using a monopolar platinum–iridium stimulating electrode (FHC) placed within the Schaffer collaterals at 0.03 Hz. Solutions containing drugs or vehicle were applied for 5 min, or as indicated. The NMDAR pore blocker MK-801 (3 μM) was applied at the end of each recording to block; MK-801 was more effective at blocking NMDARs in the presence of EU1622-240 than competitive antagonists given the increased potency for glutamate of drug-bound receptors (Ullman, Perszyk et al. 2024).

For voltage-clamp recordings of spontaneous inhibitory postsynaptic current (sIPSC), the electrodes were filled with internal solution containing (mM): 110 caesium gluconate, 4 NaCl, 30 CsCl, 5 BAPTA, 5 HEPES, 0.5 $CaCl_2$, 2 $MgCl_2$, 2 Na-ATP and 0.3 Na-GTP. The aCSF for sIPSC recording contained (mM): 130 NaCl, 2.5 KCl, 1.25 $NaH_2PO_4$, 25 $NaHCO_3$, 1 $MgCl_2$, 2 $CaCl_2$ and 20 glucose. The sIPSCs were isolated by holding the CA1 pyramidal cell at the EPSC reversal potential, which was +10 mV for our solutions. The experimental recording protocol for EPSCs and sIPSC recording was 5 min of baseline recording, followed by wash-in of 3 μM EU1622-240 for 10 min. Recordings were finished by switching to aCSF that contained EU1622-240 plus 10 μM gabazine to confirm that the currents we recorded were mediated by $GABA_A$ receptors. The last 2 min of the recording for each condition were used for data analysis. Parallel control experiments were completed with the same recording protocol, perfusing 400 μM D,L-APV in the aCSF to competitively antagonize GluN2D-containing NMDARs throughout the entire recording period.

To record a compound excitatory postsynaptic potential (EPSP) and inhibitory postsynaptic potential (IPSP), a CA1 pyramidal cell was patched in current-clamp mode with injection of current needed to set the RMP to −60 mV. The cell was rejected if the holding current was larger than ± 75 pA. Recording solution was the same aCSF as described above, but with 1.5 mM $Mg^{2+}$ and 1.5 mM $Ca^{2+}$. Electrical stimulation was applied to the Schaffer collaterals to evoke a compound postsynaptic potential, in which the depolarizing phase was classified as the EPSP and the hyperpolarizing phase as the IPSP. Electrical stimulation sites were set for two places: the distal CA3 region, ∼800 μm away from the patched CA1 pyramidal cell, or the local CA1 region, ∼200 μm away from the patched CA1 pyramidal cell (Dingledine, Roth et al. 1987). Stable recordings were made for ≥ 5 min of baseline, then 3 μM EU1622-240 was applied for ≥ 10 min. The last 2 min of recording was used for data analysis and presentation of representative traces. Amplitudes of the EPSP and IPSP were measured, and the EPSP/IPSP amplitude ratio was calculated. Given the strong potentiating effect of EU1622-240 on NMDARs, the stimulus strength was adjusted such that the initial postsynaptic potential depolarization during baseline was ∼2 mV to avoid generating action potentials during EU1622-240 application.

Series resistance was monitored throughout experiments and was typically 8−12 MΩ. For current-clamp experiments, cells were briefly switched to voltage clamp and held at −60 mV while a series of 50 ms duration, 5 mV square hyperpolarizing potentials were applied to the cell. Current-clamp recordings had series resistance measurements made at the beginning and at the end of each experiment, which usually lasted 5−10 min. All series resistances were measured off-line by analysing the peak of the capacitive charging spike and applying Ohm's law. If the series resistance ($R_s$) changed by >25% during the experiment or exceeded 30 MΩ, the cell was excluded. For voltage-clamp recordings of EPSCs, a −5 mV hyperpolarizing pulse was applied during each recording epoch before the evoked EPSC to determine $R_s$. Only stable recordings were accepted, defined as those with a drift in $R_s$ of <25% over the course of the recordings.

## Field potential recording

Wild-type postnatal day 20–25 C57BL/6J mice of both sexes were used for field potential recordings. Coronal brain slices were sectioned at 500 μm thickness using the same procedures described above. The protocol for recording and analysing field potentials corresponding to field EPSPs within the stratum radiatum close to the border of the stratum pyramidale was described by Lundbye, Toft et al. (2018). Hippocampal slices were transferred to an interface-style recording chamber (BSC 1-1, Scientific Systems Design Inc.) and bathed in the recording aCSF described above with 2.0 mM $MgSO_4$ and 3.0 mM $CaCl_2$, heated to 31–33°C. A monopolar platinum electrode was used to stimulate Schaffer collaterals at 0.03 Hz, and a recording electrode was placed in the upper one-third of the stratum radiatum of CA1. Recording electrodes were made from borosilicate glass (1.50 mm outer diameter, 1.12 mm inner diameter; World Precision Instruments) via a micropipette puller (P-1000; Sutter Instrument); pipette resistances were 4–6 MΩ, filled with recording aCSF. Signals were obtained by an A-M Systems Model 1800 field recording amplifier using ×1000 gain, with high- and low-pass filtering at 1 Hz and 5 kHz (−3 dB Bessel), respectively. Recordings were digitized at 20 kHz using NacGather acquisition software v.2.07 (Theta Burst, Irvine, CA, USA).

For input–output (I/O) curves, the stimulation level was ∼40–300 μA at 0.1 ms duration. Three sweeps were collected for each stimulation level during I/O curve recording. The slices that went through I/O curve recording were not used for assessment of LTP. For LTP recording, a brief preliminary I/O curve was recorded, with one sweep per stimulation level at 30, 60 and 100 μA stimulation levels, then the stimulation level was set to the level that yielded a response amplitude of ∼50% compared with that obtained in response to 100 μA stimulation. The slice was equilibrated in the recording chamber with the treatment solution for 20 min, resulting in a baseline change of <5%. Then the recording was started for 20 min of baseline, followed by a theta-burst stimulation protocol (each protocol consisted of a train of 10 theta bursts, each containing four pulses at 100

Hz with an inter-burst interval of 200 ms) to induce a form of LTP. The LTP recording protocol was: (i) 20 min low-frequency stimulation at 0.03 Hz; (ii) theta-burst stimulation; (iii) repeat periods (i) and (ii) three more times such that a total of four theta-burst stimulation protcols were applied; and (iv) finished with a 20 min low-frequency stimulation (Lundbye, Toft et al. 2018; Morgan & Teyler 2001). The effects of both vehicle (0.3% DMSO) and 3 μM EU1622-240 were tested. To avoid contamination of remaining EU1622-240 between each recording, perfusion lines, chambers and slice anchors were washed once with 50 ml of 10% DMSO, then twice with 50 ml of 70% ethanol, twice with 50 ml of deionized $H_2O$, then once with 50 ml of external recording solution after each slice.

### Immunohistochemistry

*Post hoc* immunohistochemical staining for biocytin was performed to confirm that the proper cell type was recorded, used largely to confirm that recordings made from cells in *stratum pyramidale* were from pyramidal cells (i.e. contained a large apical dendrite with dendritic spines) and not from GABAergic interneurons (Pelkey, Chittajallu et al. 2017). After each recording, the patch pipette was withdrawn carefully from the patched cell to allow recovery for resealing of the cell membrane. The slice was subsequently transferred into 4% paraformaldehyde and stored at 4°C overnight. The next day, the slice was washed three times with PBS, then incubated in a 1.2% Triton-X–PBS solution for 10 min. The slice was subsequently incubated in the following solution overnight at room temperature for biocytin staining: 10% normal goat serum (Abcam, ab7481), 0.3% Triton-X (Fisher Scientific, BP151-500), 2% bovine serum albumin (Jackson ImmunoResearch, 156 231), and anti-streptavidin Alexa546 1:500 (Invitrogen, S11225) dissolved in $1\times$ PBS. On the next day, the slice was washed with PBS three times, mounted onto a slide, and allowed to air dry for 4 h. Then, a coverslip was placed over the slide using Prolong Gold mounting medium (Invitrogen, P36934). Neurons were imaged at 20x magnification in 1-μm-thick optical sections on a spinning disc confocal microscope, and a maximum-intensity projection was made from the *z*-stack.

### Dissociated hippocampal cell culture

For live cell $Ca^{2+}$-imaging experiments, dissociated hippocampal neurons were prepared from male and female postnatal day 0–2 Sprague–Dawley rat pups. Pups were rapidly decapitated and the hippocampi dissected and dissociated using papain (20 units/ml) digestion for 45–60 min at room temperature with gentle rocking. The cells were triturated five or six times through a fire-polished pipette and plated on 18 mm glass coverslips coated with poly-D-lysine (Sigma, P6407) in minimal essential medium containing 10% FBS (Cytiva, SH30071.02HI) and penicillin–streptomycin, at a density of 100,pp000 cells per well (12-well dish). After 24 h, the medium was switched to Neurobasal-A medium (Invitrogen, 10888022) with B27 (Invitrogen, 17504044) and Glutamax (ThermoFisher, 35050061) with anti-mitotic (10 μM 5-fluorodeoxyuridine) added after 7 days in culture to block glial overgrowth. The neurons were maintained at 37°C in a humidified incubator with 5% $CO_2$. Hippocampal neurons were transfected between 14 and 16 days *in vitro* using lipofectamine 2000 (ThermoFisher, 11668027) according to the manufacturer's protocol. Microscopy experiments were performed within 24 h of transfection.

### NMDAR $Ca^{2+}$ imaging and analysis

Live-cell $Ca^{2+}$ imaging was performed at 32°C on an Olympus IX-71 equipped with a spinning disc scan head (Yokogawa). Images were acquired using an Olympus $60\times$ Plan Apochromat 1.4 numerical aperture objective and collected on a $1024 \times 1024$ pixel Andor iXon EM-CCD camera, with pixels binned $3 \times 3$. Data acquisition and analysis were performed with MetaMorph (Molecular Devices) and ImageJ software. To image NMDAR-mediated quantal $Ca^{2+}$ transients, neurons were transfected with GCaMP6s (Addgene plasmid #40753) at 12–14 days *in vitro*. Neurons were imaged 1 day later in $Mg^{2+}$-free extracellular solution containing (mM): 130 NaCl, 5 KCl, 10 HEPES, 30 glucose, 2 $CaCl_2$, 0.003 glycine and 0.002 TTX, pH 7.4. Single *z*-plane images of the dendritic arbor were acquired at 15 Hz for 1–2 min to record baseline quantal $Ca^{2+}$ transients. Then, vehicle (DMSO) or EU1622-240 was added to the recording chamber (for final concentrations of 1, 3 or 9 μM), and the same cells were imaged 2–10 min after drug application. We performed a third round of imaging of the same cells after addition of 100 μM D,L-APV and 20 μM MK-801 (in the continued presence of EU-1622-240) to confirm that enhanced $Ca^{2+}$ entry was NMDAR dependent.

To quantify the quantal $Ca^{2+}$ transient amplitude, regions of interest were drawn around clearly resolved spines in the baseline (prior to EU1622-240 addition, see Supporting_Video S1). The same regions of interest were transferred to the post-drug (see Supporting_Video S1). The mean background-subtracted GCaMP6s fluorescence within each region of interest was determined pre- and post-drug. Only synapses displaying $Ca^{2+}$ transient events both before and after EU1622-240 addition were included in the analysis, allowing paired comparison of event

amplitudes at each individual synapse, pre/post drug application.

## Data analysis

Whole-cell electrophysiology data were collected by Clampex 10.7 with MultiClamp 700B Commander. The sIPSC data were analysed offline using MiniAnalysis 6 (Synaptosoft) with an 8-pA threshold for event detection, which was 5-fold higher than our peak root mean square noise. For all whole-cell current responses, all data were analysed offline using ChanneLab 2.210604 (Dr Stephen F. Traynelis) or pClamp 10.3 (Molecular Devices) on the average response per condition, except for charge transfer for evoked NMDAR-mediated EPSCs, for which each trace was analysed individually. Rise time was measured as the 10%–90% fractional response of the rising phase of each current response. Peak amplitude was determined by finding the maximum of a moving search average of three to five data points. The deactivation time course for evoked NMDAR-mediated EPSCs was measured by fitting a dual-exponential function to the decay; the time constants are reported as weighted deactivation time constants ($\tau_w$), calculated using the following equation ($\tau_w$, weighted deacivation time constants; $Amp_{fast}$, amplitude of fast component of the EPSC; $Amp_{slow}$, amplitude of slow component of the EPSC; $\tau_{fast}$, deacivation time constants of fast component of the EPSC; $\tau_{slow}$, deacivation time constants of slow component of the EPSC):

$$\tau_w = \left[ Amp_{fast} / (Amp_{fast} + Amp_{slow}) \right]$$
$$\tau_{fast} + \left[ Amp_{slow} / (Amp_{fast} + Amp_{slow}) \right] \tau_{slow})$$

Charge transfer of current responses was calculated as the area under the curve (amplitude $\times$ $\tau$) for each response. The EPSP/IPSP amplitudes were analysed offline by Clampfit 10.3. Traces with action potentials or postsynaptic potentials that were not able to be separated from the stimulation artefact were rejected. For input resistance, we analysed the response to $-30$ pA, 200 ms hyperpolarizing injected current to obtain the membrane voltage change in response to current injection. All other input resistance data were analysed by fitting a linear regression of the membrane potential response to step current injection. Input resistance was calculated according to Ohm's law. Slice field potential data were analysed offline by NACShow2 (Theta Burst Corp.) to obtain the slope of the rising phase of the field EPSP (fEPSP).

Data analysis and figures were made with the assistance of GraphPad Prism 9.3.1 (GraphPad Software, LLC) and OriginPro 9.0.0 (OriginLab Corporation). Data are presented as the mean $\pm$ SD. Significance was determined via Student's *t*-test or ANOVA as appropriate.

An $\alpha$-value of $p < 0.05$ was accepted as statistically significant. Evoked EPSCs were analysed by Student's paired *t*-test (two-tailed); current-clamp data were analysed by ordinary one-way ANOVA with Dunnett's multiple comparisons test, and bicuculline-treated intrinsic membrane properties were analysed by repeated-measures ANOVA with Tukey's multiple comparisons test (details can be found in each table). When multiple tests were performed on the same recording, $\alpha$-values were corrected to prevent family-wise errors using the Bonferroni *post hoc* correction method. All studies were designed such that an effect size of at least one was detected at $\geq 80\%$ power.

## Results

To determine the actions on neuronal function of the PAM EU1622-240, we designed a series of experiments to monitor its actions on evoked NMDAR-mediated EPSCs in both pyramidal cells and GABAergic interneurons, inhibitory tone, calcium transients in cultured neurons, and hippocampal synaptic plasticity. This PAM can potentiate recombinant NMDARs with $EC_{50}$ values of 0.6, 1.0 and 1.1 µM for GluN1/GluN2B, GluN1/GluN2C and GluN1/GluN2D, respectively, and produces a maximal potentiation of 390, 720 and 830% of control, respectively (Chou, Epstein et al. 2024; Fritzemeier, Akins et al. 2025; Ullman, Perszyk et al. 2024). From these concentration–response curves, we predict that concentrations of 0.19–0.46 µM EU1622-240 will double the current response at steady state. Figure 1 illustrates the position and morphology of typical CA1 pyramidal cells and CA1 interneurons included in this study, in addition to the stimulation and recording sites.

### EU1622-240 enhances interneuron NMDAR-mediated EPSCs

Initially, we assessed the effects of EU1622-240 on evoked synaptic NMDAR-mediated EPSCs from both CA1 inter-neurons and CA1 pyramidal cells. We held neurons at $+40$ mV to remove voltage-dependent $Mg^{2+}$-induced channel block, and we evoked EPSCs with Schaffer collateral stimulation. Figure 2 shows the effects of 3 µM EU1622-240 on evoked EPSCs recorded from CA1 stratum radiatum interneurons (Fig. 2*A*) and pyramidal cells (Fig. 2*B*). For both CA1 interneurons and pyramidal cells, we observed a modest increase in EPSC amplitude and charge transfer (Fig. 2*C–E*; Table 1). Interneurons showed no detectable increase in the mean weighted time constant for deactivation (Fig. 2*D1*; Table 1), and no significant effect for decay time was observed for CA1 pyramidal cells (Fig. 2*D2*; Table 1).

## EU1622-240 enhances interneuron excitability

We tested the impact of EU1622-240 (labelled as 'EU240' in blue in Fig. 3) on neuronal function recorded under current clamp from hippocampal CA1 stratum radiatum interneurons and pyramidal cells (Fig. 3). We washed in sequentially increasing concentrations of EU1622-240 of 0.1, 0.3, 1, 3 and 6 µM or matched vehicle (DMSO). Each drug was washed in for 5 min, with the first 1 min estimated to be the time needed to reach the bath (24 s) plus the time it takes to exchange the bath volume five times (38 s). Throughout the recording, we monitored instantaneous firing frequency and RMP. Figure 3A1 shows a typical recording from a CA1 stratum radiatum interneuron during a baseline period, followed by the application of increasing

concentrations of EU1622-240. There is a clear increase in the action potential firing frequency (Fig. 3A1) in this interneuron, in addition to a depolarization of RMP (Fig. 3A2) during drug application. Figure 3B shows that vehicle (0.01–0.6% DMSO) had no effect over the same recording period. Figure 4A–C summarizes the instantaneous firing frequency, RMP and input resistance for interneurons during the application of EU1622-240. These data support the idea that EU1622-240 can increase the excitability of interneurons when applied at concentrations of 3–6 µM. The concentration of 3–6 µM is likely to correspond to a 'within-the-slice' drug concentration of ≥1 µM, assuming the drug takes several minutes to penetrate the tissue. We also observed both statistically significant depolarization at 6 µM and reduction of input resistance at 3 µM (see Table 2).

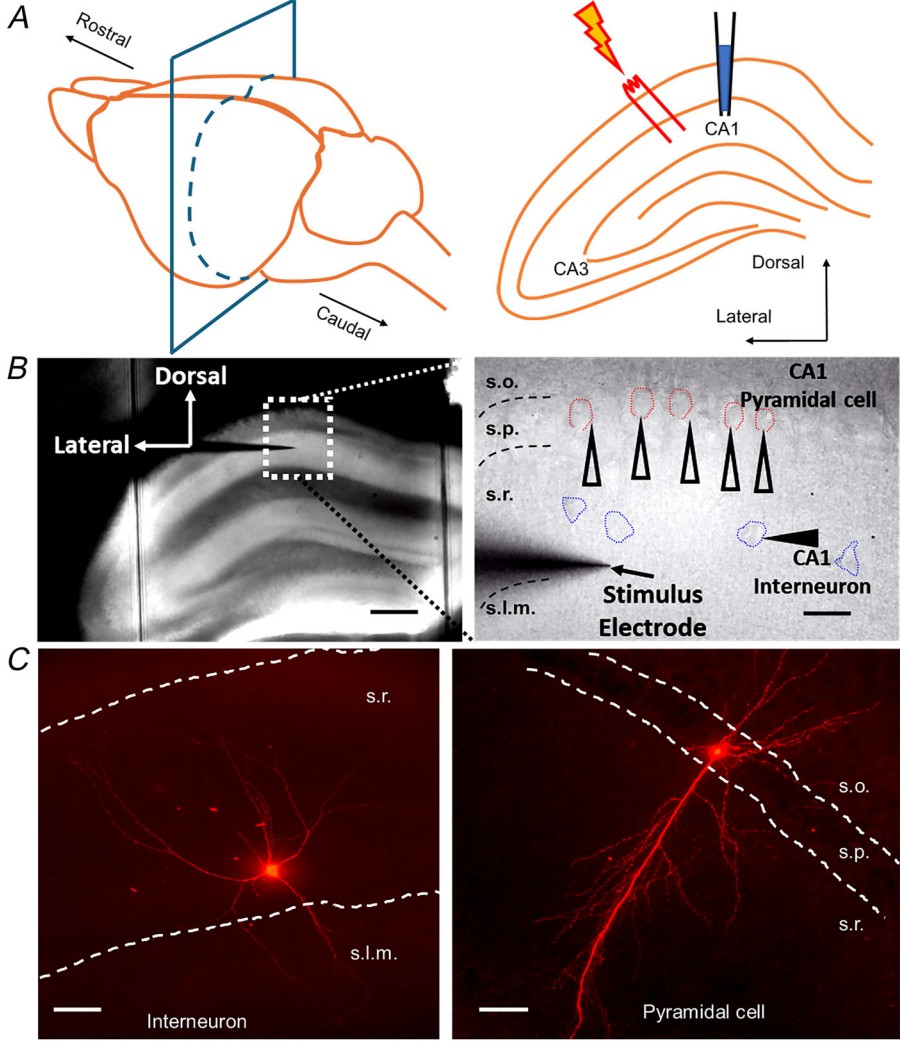

**Figure 1. Identification of CA1 pyramidal cells and interneurons in acute hippocampal slices**
*A* and *B*, coronal sections of mouse brain and identification of stimulus/recording sites. *B*, open arrowheads indicate the cell body of pyramidal cells (with red dotted outline), and black arrowhead indicates an interneuron (with blue dotted outline). Scale bars: 1 mm in left image; 250 µm in right image. *C*, maximum-intensity projection image of a biocytin-filled interneuron (left) and pyramidal cell (right). Scale bar: 250 µm.

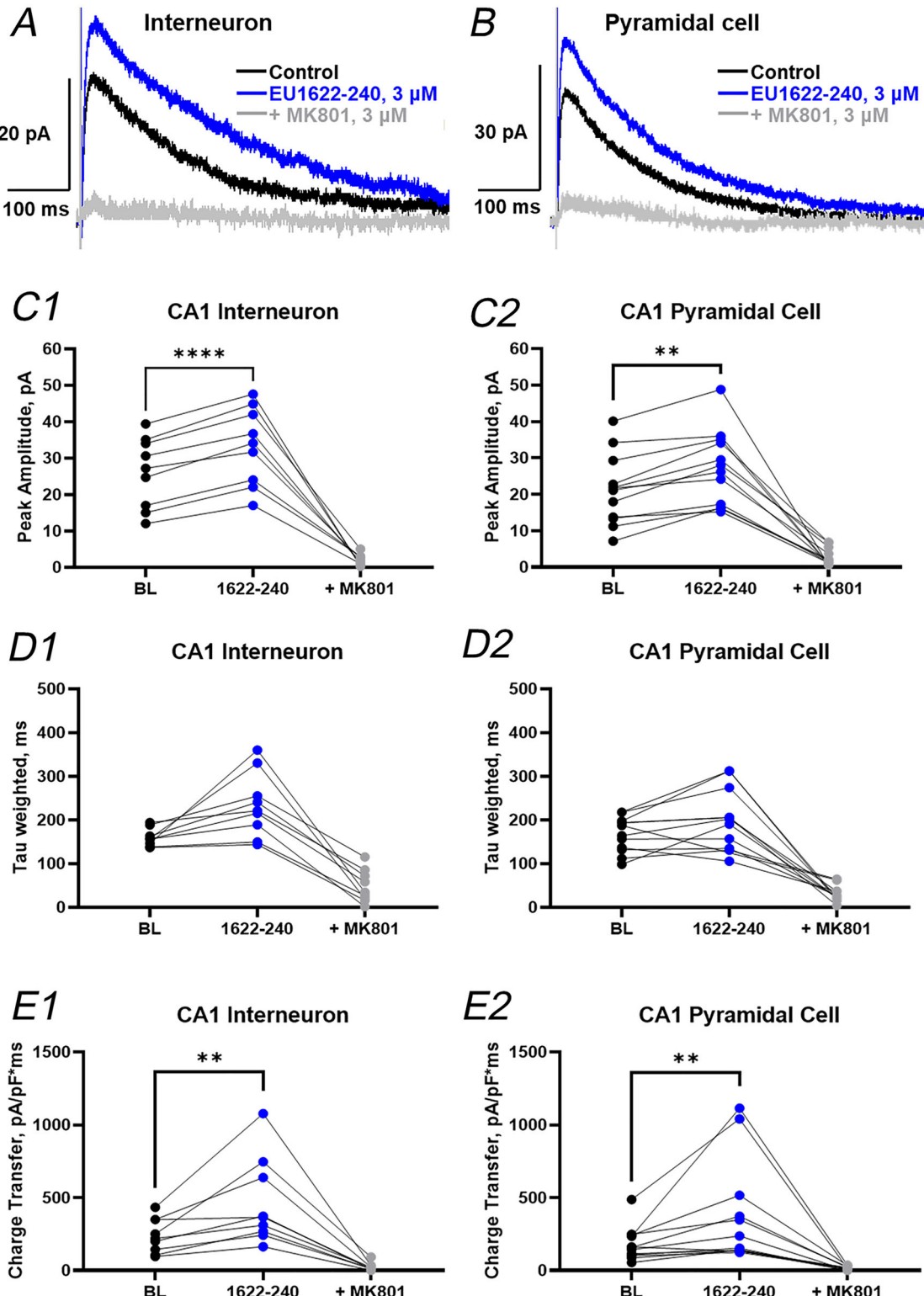

**Figure 2. EU1622-240 alters the NMDAR-mediated component of the evoked EPSC**
*A* and *B*, representative evoked synaptic currents are shown for a CA1 interneuron (*A*) and a CA1 pyramidal cell (*B*) recorded at a holding potential of +40 mV before (black) and after the application of 3 µM EU1622-240 (blue). MK801 was added at the conclusion of the recordings to confirm that the current was mediated by NMDARs (grey). *C–E*, NMDAR peak amplitude (*C1* and *C2*), weighted time course of deactivation (*D1* and *D2*) and mean charge transfers (*E1* and *E2*) are given for interneurons and pyramidal cells, respectively. \*\**P* ≤ 0.01 and \*\*\**P* ≤ 0.0001, by Student's paired *t* test. Data are from nine interneurons and 12 pyramidal cells.

**Table 1. The effect of EU1622-240 on evoked NMDAR-mediated EPSCs**

| Parameter | Interneuron, $n = 9$ | | | Pyramidal cell, $n = 12$ | | |
|---|---|---|---|---|---|---|
| | Control | EU1622-240 | $P$-value | Control | EU1622-240 | $P$-value |
| Peak amplitude (pA) | $26 \pm 12$ | $33 \pm 14$ | $<0.0001$ | $21 \pm 9.6$ | $27 \pm 10$ | 0.00284 |
| Rise time (ms) | $11 \pm 5.4$ | $7.6 \pm 4.5$ | – | $8.0 \pm 3.6$ | $8.5 \pm 4.0$ | – |
| $\tau_{FAST}$ (ms) | $119 \pm 36$ | $155 \pm 46$ | – | $138 \pm 50.3$ | $152 \pm 52$ | – |
| $\tau_{slow}$ (ms) | $430 \pm 239$ | $604 \pm 399$ | – | $233 \pm 224$ | $529 \pm 521$ | – |
| % $\tau_{fast}$ | 65% | 65% | – | 53% | 53% | – |
| $\tau_{w}$ (ms) | $160 \pm 64$ | $234 \pm 167$ | 0.0200 | $167 \pm 70$ | $246 \pm 163$ | 0.0823 |
| Charge transfer (ms pA/pF) | $239 \pm 131$ | $465 \pm 306$ | 0.00159 | $178 \pm 140$ | $371 \pm 353$ | 0.00901 |

Data are presented as the mean $\pm$ SD. Statistical analysis: Student's paired $t$ test with Bonferroni family-wise correction. $\tau_{w}$, weighted deacivation time constants; $\tau_{fast}$, deacivation time constants of fast component of EPSC; $\tau_{slow}$, deacivation time constants of slow component of EPSC; % $\tau_{fast}$, percentage of $\tau_{fast}$ within $\tau_{fast} + \tau_{slow}$; pA, picoamp; ms, millisecond; pF, picofarad.

**Table 2. Intrinsic properties of CA1 interneurons and pyramidal cells in response to EU1622-240**

| [EU1622−240] (µM) | Interneuron, $n = 11$ | | | Pyramidal cell, $n = 10$ | | |
|---|---|---|---|---|---|---|
| | Spontaneous frequency (Hz) | RMP (mV) | $R_{input}$ (M$\Omega$) | Spontaneous frequency (Hz) | RMP (mV) | $R_{input}$ (M$\Omega$) |
| Baseline | 0 | $-62 \pm 3.2$ | $210 \pm 36$ | $0.3 \pm 1.0$ | $-61 \pm 3.7$ | $204 \pm 35$ |
| 0.1 | $0.20 \pm 0.5$ | $-63 \pm 4.4$ | $208 \pm 43$ | $0.2 \pm 0.5$ | $-61 \pm 2.2$ | $224 \pm 69$ |
| 0.3 | $0.06 \pm 0.2$ | $-63 \pm 4.4$ | $182 \pm 30$ | $0.8 \pm 1.4$ | $-61 \pm 2.3$ | $222 \pm 63$ |
| 1.0 | $0.40 \pm 1.0$ | $-62 \pm 4.8$ | $178 \pm 32$ | $1.5 \pm 2.0$ | $-62 \pm 3.6$ | $230 \pm 50$ |
| 3.0 | $4.1 \pm 1.6$ $P = 0.0073$ | $-59 \pm 4.7$ | $170 \pm 22$ $P = 0.017$ | $2.1 \pm 4.4$ | $-62 \pm 4.7$ | $235 \pm 53$ |
| 6.0 | $12 \pm 2.3$ $P = 0.00082$ | $-57 \pm 2.9$ $P = 0.027$ | $187 \pm 19$ | – | – | – |

Data are presented as the mean $\pm$ SD. The $P$-values are calculated by comparing each EU1622-240 concentration with baseline by one-way ANOVA with Dunnett's multiple comparisons test.

Abbreviations: RMP, resting membrane potential; $R_{input}$, input resistance; µM, micromolar; Hz, hertz; mV, millivolts; M$\Omega$, megaohm.

These results are consistent with an EU1622-240-induced increase in excitatory drive onto interneurons through potentiation of NMDARs that are likely to contain the GluN2D subunit, which showed the strongest level of enhancement by this PAM (Fritzemeier, Akins et al. 2025).

Figures 3C1 and C2 shows a typical recording from a CA1 pyramidal cell during a baseline period, followed by a recording during the application of increasing concentrations of EU1622-240 (from 0.1 to 6 µM) by the same protocol as used for interneurons. Figure 3D1 and D2 shows the effects of vehicle (0.01–0.6% DMSO) during the same recording period. There is no obvious change in neuronal properties monitored during either EU1622-240 or vehicle. Figure 4D and E summarizes the instantaneous firing frequency, RMP and input resistance during the application of EU1622-240 or vehicle onto CA1 pyramidal cells. EU1622-240 did not produce any detectable effects on these parameters in CA1 pyramidal cells. These data suggest that the larger effects on GluN2D-containing

receptors might underlie increased actions of EU1622-240 on interneurons than on CA1 pyramidal cells, given that most hippocampal interneurons express GluN2D (Perszyk, DiRaddo et al. 2016; von Engelhardt, Bocklisch et al. 2015).

Next, we selected 3 µM EU1622-240 as a concentration to use to explore the actions of EU1622-240 on neuronal properties. Before each recording epoch, a series of current injections was applied to assess the firing frequency for CA1 stratum radiatum interneurons and pyramidal cells. Figure 5 shows representative responses to increasing current injections for both CA1 stratum radiatum interneurons (Fig. 5A and B) and CA1 pyramidal cells (Fig. 5C and D). In the presence of EU1622-240, injection of hyperpolarizing current failed to suppress firing (Fig. 5A2), whereas injection of depolarizing current produced more depolarization and spike firing in CA1 interneurons (Fig. 5B2) than in CA1 pyramidal cells (Fig. 5C2 and D2). Figure 5E–H

plots the change in membrane potential for current injection for interneurons and CA1 pyramidal cells. These data suggest that EU1622-240 decreases input resistance for CA1 interneurons [Fig. 5E; $P = 0.0209$, $F(1, 20) = 6.288$, two-way ANOVA with Bonferroni's multiple comparisons test], given that the current injections produce less depolarization, a result consistent with an EU1622-240-induced increase in the open probability of NMDARs that contain the GluN2D subunit. There is no clear effect of EU1622-240 on this same relationship determined for CA1 pyramidal cells [Fig. 5G, $P = 0.092$, $F(1, 23) = 3.089$, two-way ANOVA with Bonferroni's

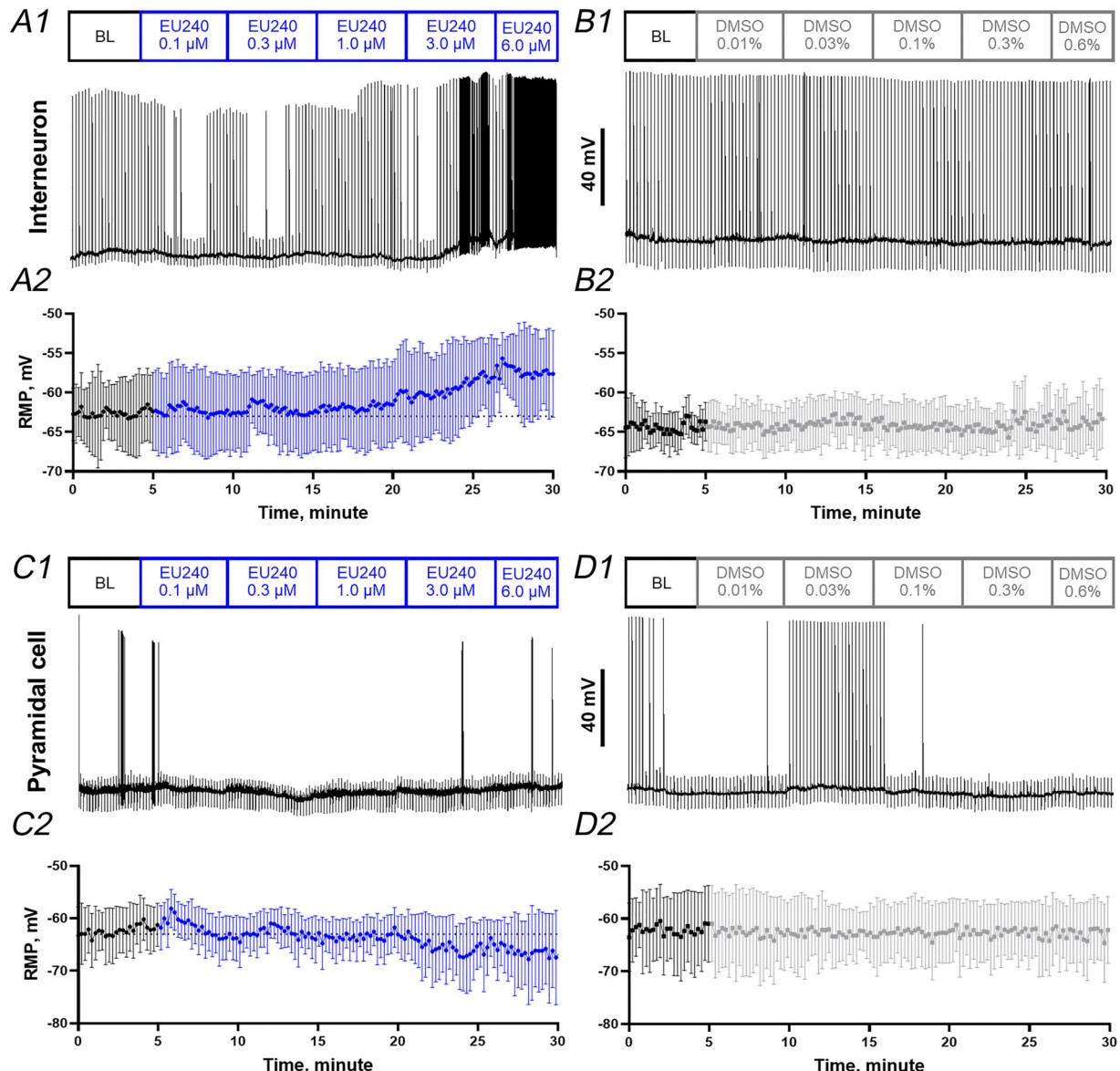

**Figure 3. EU1622-240 depolarizes hippocampal CA1 interneurons and increases spike firing, with minimal effects on hippocampal CA1 pyramidal cells**

*A1* and *B1*, representative current-clamp recording from a CA1 stratum radiatum interneuron during application of increasing concentrations of either EU1622-240 (blue; *A1*) or vehicle (grey; *B1*). Each concentration given for EU1622-240 was applied for 5 min before switching to the next drug concentration. A −30 pA 200 ms hyper-polarizing current injection was administered intermittently throughout the recording to assess input resistance. *A2* and *B2*, Composite RMP data are shown for increasing concentrations of EU1622-240 (*n* = 11 cells; *A2*) or vehicle (*n* = 7 cells; *B2*). *C1* and *D1*, representative current-clamp recording from a CA1 pyramidal cell during application of increasing concentrations of either EU1622-240 (blue; *C1*) or vehicle (grey; *D1*). A brief hyperpolarizing current injection was administered throughout the recording to assess input resistance. *C2* and *D2*, composite RMP data are shown for increasing concentrations of EU1622-240 (*n* = 10 cells; *C2*) or vehicle (*n* = 7 cells; *D2*). All data are the mean ± SD. Abbreviations: BL, baseline; EU240, EU1622-240; RMP, resting membrane potential.

multiple comparisons test]. This result is consistent with that described in the recordings shown in Fig. 4.

There is also an increase in action potential firing in response to depolarizing current injection for CA1 stratum radiatum interneurons [Fig. 5*F*; $P = 0.0248$, $F(1, 10) = 6.963$, two-way ANOVA with Bonferroni's multiple comparisons test]. In CA1 pyramidal cells, there is a significant increase in depolarization-induced firing frequency after EU1622-240 treatment when compared with baseline [Fig. 5*H*; $P = 0.028$, $F(1, 8) = 7.177$, two-way ANOVA with Bonferroni's multiple comparisons test]. These data show an increased response to depolarizing current injection, indicating that EU1622-240 can increase the intrinsic excitability of both CA1 interneurons and pyramidal cells.

To confirm that the expression of GluN2D-containing NMDARs is involved in the differential effects of EU1622-240 on neuronal activity of CA1 interneurons and pyramidal cells, we repeated current-clamp recordings with the same protocol on brain slices from *Grin2D* knockout (GluN2D-KO) mice. Figure 6*A* shows a representative current-clamp recording, with baseline, 10 min of 3 μM EU1622-240 wash-in, and co-administration of 3 μM MK-801 with 3 μM EU1622-240. At 3 μM, EU1622-240 did not alter input resistance, RMP and depolarization-evoked firing

**Table 3. GluN2D-KO CA1 interneuron intrinsic membrane properties are not affected by EU1622-240**

| Conditions | Interneuron | | |
|---|---|---|---|
| | RMP (mV) | $R_{input}$ (MΩ) | $n$ |
| Baseline | $-76 \pm 8.0$ | $198 \pm 61$ | 9 |
| 3 μM EU1622-240 | $-78 \pm 8.4$ | $197 \pm 81$ | 9 |
| 3 μM EU1622-240 + MK801 | $-79 \pm 9.0$ | $235 \pm 82$ | 9 |

Data are presented as the mean ± SD. The *P*-value is calculated by one-way ANOVA with Tukey's multiple comparisons test; no significant differences were found for RMP (resting membrane potential) or $R_{input}$ (input resistance). Abbreviations: RMP, resting membrane potential; R$_{input}$, input resistance; μM: micromole; mV: millivolts; MΩ: megaohm.

frequency in CA1 interneurons from GluN2D-KO mice (Table 3; Fig. 6*B–F*).

Besides GluN2D, EU1622-240 also potentiates GluN2B, which is expressed on both CA1 stratum radiatum interneurons and pyramidal cells (Booker, Sumera et al. 2021; Hansen, Ogden et al. 2014). We hypothesized that there should be a change in the spontaneous excitatory signalling onto CA1 pyramidal cells via the effect of EU1622-240 on NMDARs. To evaluate whether

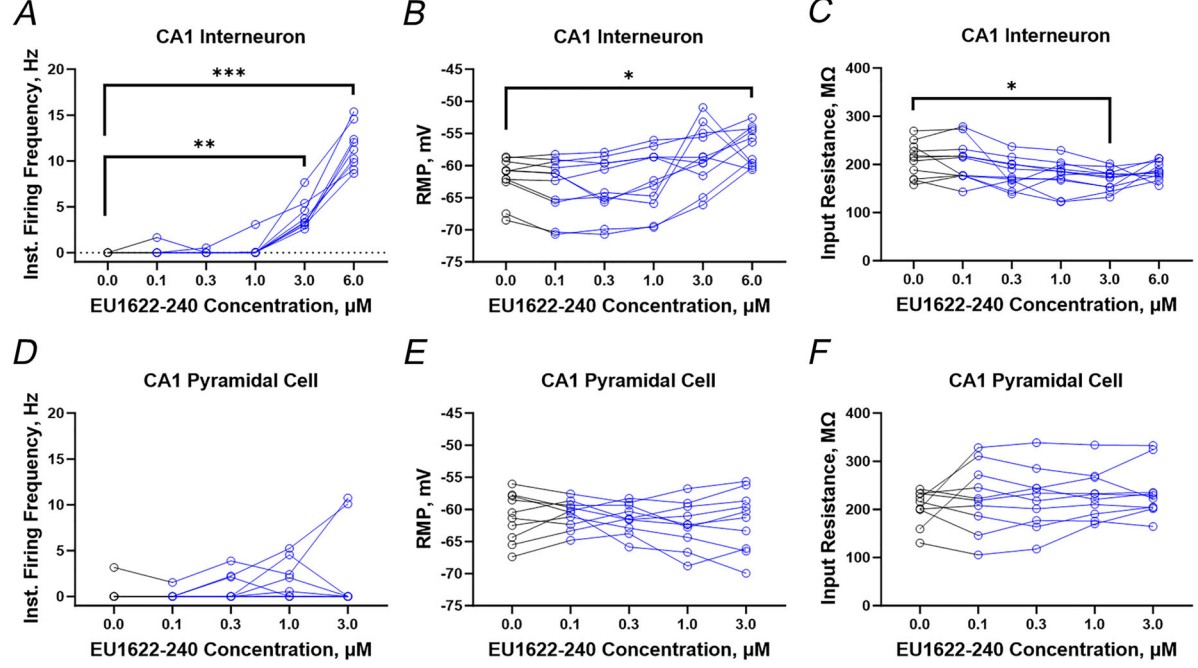

**Figure 4. Effects of EU1622-240 on CA1 pyramidal cell and interneuron properties**
The mean instantaneous firing frequency (*A* and *D*), resting membrane potential (RMP; *B* and *E*), and input resistance (*C* and *F*) are shown for CA1 interneurons (*A–C*) and pyramidal cells (*D–F*) during application of increasing concentrations of EU1622-240. Baseline values were recorded over 5 min prior to the start of drug application. Interneuron, $n = 11$ cells; *$P \leq 0.05$, **$P \leq 0.01$ and ***$P \leq 0.001$, by one-way ANOVA with Dunnett's multiple comparison test. Pyramidal cell, $n = 10$ cells; no significant changes were detected by ordinary one-way ANOVA with Dunnett's multiple comparison test.

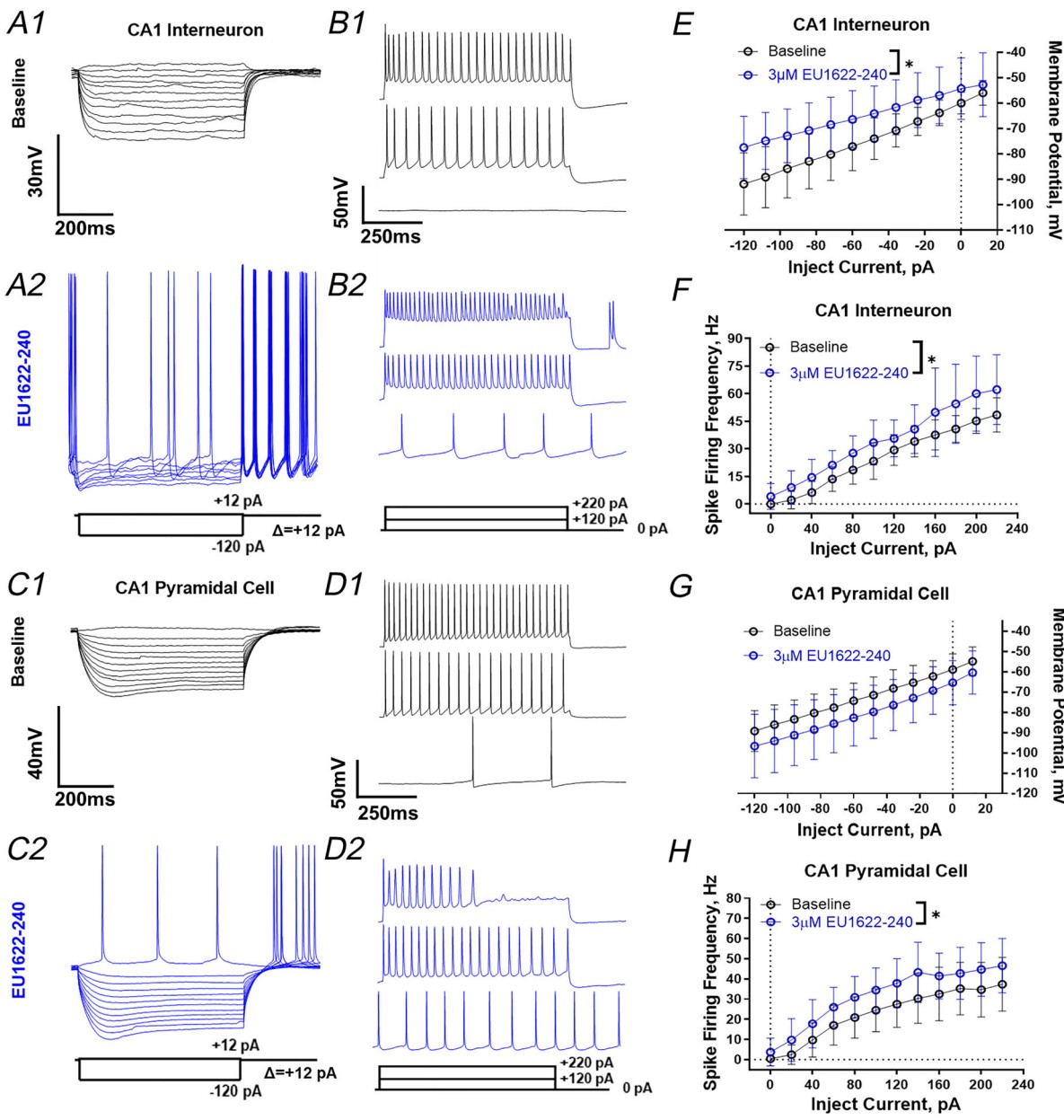

**Figure 5. At 3 μM, EU1622-240 alters CA1 interneuron intrinsic properties, but has minimal effects on CA1 pyramidal cell intrinsic properties**

*A*, the response is shown for a representative CA1 interneuron to hyperpolarizing current injections from −120 pA, in 10 pA steps, in vehicle (0.3% DMSO; *A1*) or 3 μM EU1622-240 (*A2*). *B*, expanded recordings are shown during 0, +120 and +220 pA depolarizing current injections from the interneuron in *A*, either before (*B1*) or after EU1622-240 treatment (*B2*). *C*, the response is shown for a representative CA1 pyramidal cell to hyperpolarizing current injections to −120 pA, in 10 pA steps, in vehicle (0.3% DMSO; *C1*) or 3 μM EU1622-240 (*C2*). *D*, expanded recordings are shown during 0, +120 and +220 pA depolarizing current injection from the pyramidal cell in *C*, either before (*D1*) or after EU1622-240 treatment (*D2*). *E* and *F*, the mean membrane potential is plotted during hyperpolarizing current injection into interneurons (*E*) or pyramidal cells (*F*). *G* and *H*, the mean firing frequency is plotted in response to depolarizing current injection into interneurons (*G*) or pyramidal cells (*H*). All data are the mean ± SD. *$P \leq 0.05$, two-way ANOVA with Bonferroni's multiple comparisons test.

EU1622-240 potentiated NMDAR function in response to spontaneous synaptic glutamate release, we imaged dissociated hippocampal neurons transfected with the genetically encoded $Ca^{2+}$ indicator GCaMP6s (Chen, Wardill et al. 2013) in $Mg^{2+}$-free aCSF with TTX to block evoked neurotransmitter release (Fig. 7*A*). In these conditions, brief $Ca^{2+}$ transients corresponding to NMDAR activation by spontaneous vesicle release events (quantal $Ca^{2+}$ transients) can be visualized at individual dendritic spines (Reese & Kavalali 2016; Sinnen, Bowen et al. 2016) (Fig. 7*A*). Supporting Video S1 demonstrates the $Ca^{2+}$ imaging of one neuron before EU1622-240 treatment (left panel, pre-EU1622-240), with the $Ca^{2+}$ imaging of the same neuron following administration of 3 μM EU1622-240 (right panel, post-EU1622-240). This assay allows us to compare NMDAR function directly at the same synapses before and after EU1622-240 application (Fig. 7*B*). As expected, we observed a robust increase in quantal $Ca^{2+}$ transient amplitude following application of 3 μM EU1622-240, with a similar increase in the integrated $Ca^{2+}$ signal, which reflects quantal $Ca^{2+}$ transient amplitude, kinetics and frequency over

the duration of the imaging session (Fig. 7*C–F*). Similar results were obtained for 9 μM EU1622-240, whereas 1 μM did not appear to potentiate NMDAR $Ca^{2+}$ entry significantly (Fig. 7*G*). Blockade of NMDARs by D,L-APV and MK-801 confirms that the effect of EU1622-240 on quantal $Ca^{2+}$ transients was NMDAR dependent (Fig. 7*B* and *C*; Supporting Video S2).

To investigate whether there is an increase in inhibitory tone and whether the increased inhibitory signal might affect intrinsic membrane properties, we repeated the current-clamp experiment in CA1 pyramidal cells in the presence of bicuculline to block $GABA_A$ receptor-mediated signalling. We detected both depolarization of RMP and increased spontaneous firing in CA1 pyramidal cells with co-administration of bicuculline and EU1622-240 (Fig. 8*A–D*). The bicuculline treatment alone did not affect the basal firing frequency of CA1 pyramidal cells even with a moderate but significantly depolarized RMP (Fig. 8*E*). Co-application of EU1622-240 with bicuculline produced a clear depolarization of CA1 pyramidal cells, which led to an increased firing frequency (see Table 4).

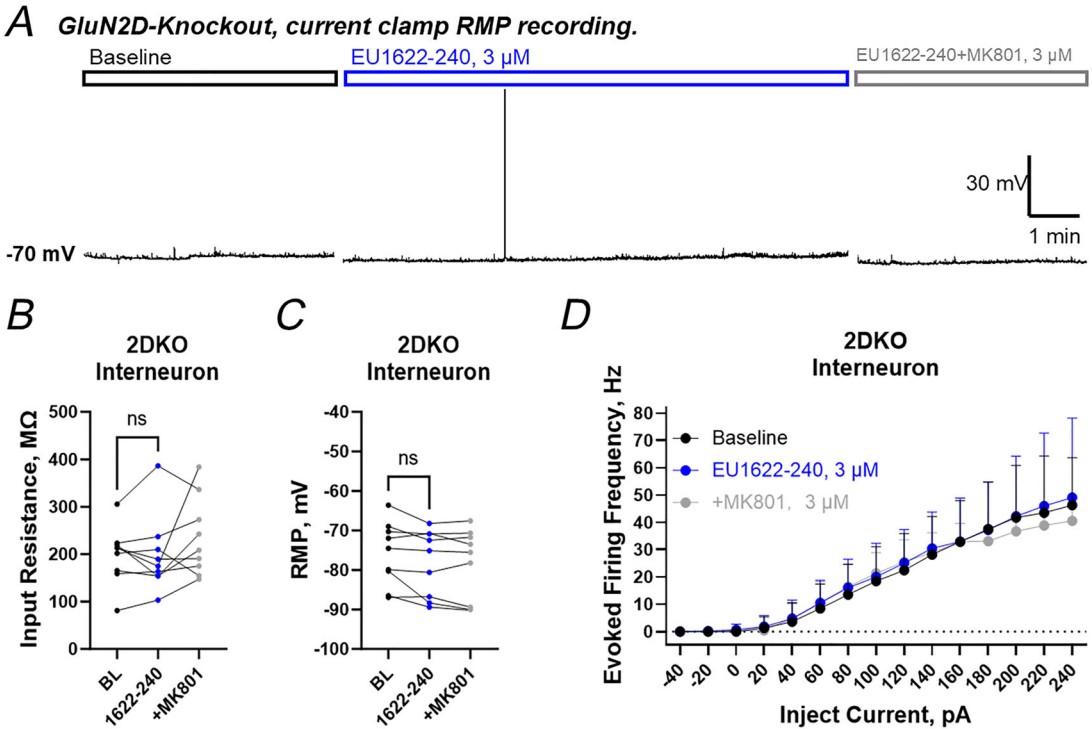

**Figure 6. *GluN2D* knockout eliminates the effects of 3 μM EU1622-240 on CA1 interneuron excitability**
*A*, representative current recordings obtained from a CA1 stratum radiatum interneuron from a *GluN2D* knockout mouse showing baseline (black outlined bar region), wash-in of 3 μM EU1622-240 for 10 min (blue outlined bar region) and co-application of 3 μM MK801 with 3 μM EU1622-240 (grey outlined bar region). *B* and *C*, no change was detected in the input resistance (*B*) or RMP (*C*) in response to 3 μM EU1622-240 in slices from *GluN2D* knockout mice. *D*, firing frequency was unchanged in the presence of 3 μM EU1622-240. All data are the mean ± SD. *n* = 9 cells. Statistical significance was determined by Student's paired *t* test in *B* and *C*. Abbreviations: ns, not significant; RMP, resting membrane potential; 2DKO, *GluN2D* knockout.

One potential interpretation is that the increased tonic inhibition produced by enhanced NMDAR drive onto interneurons masks a depolarizing effect of EU1622-240 on CA1 pyramidal cells, which occurs via potentiation of GluN2B-containing NMDARs on pyramidal cells.

## EU1622-240 enhanced interneuron excitability increases inhibitory tone onto CA1 pyramidal cells

The increased excitability and increased NMDAR-mediated synaptic drive onto CA1 inter-

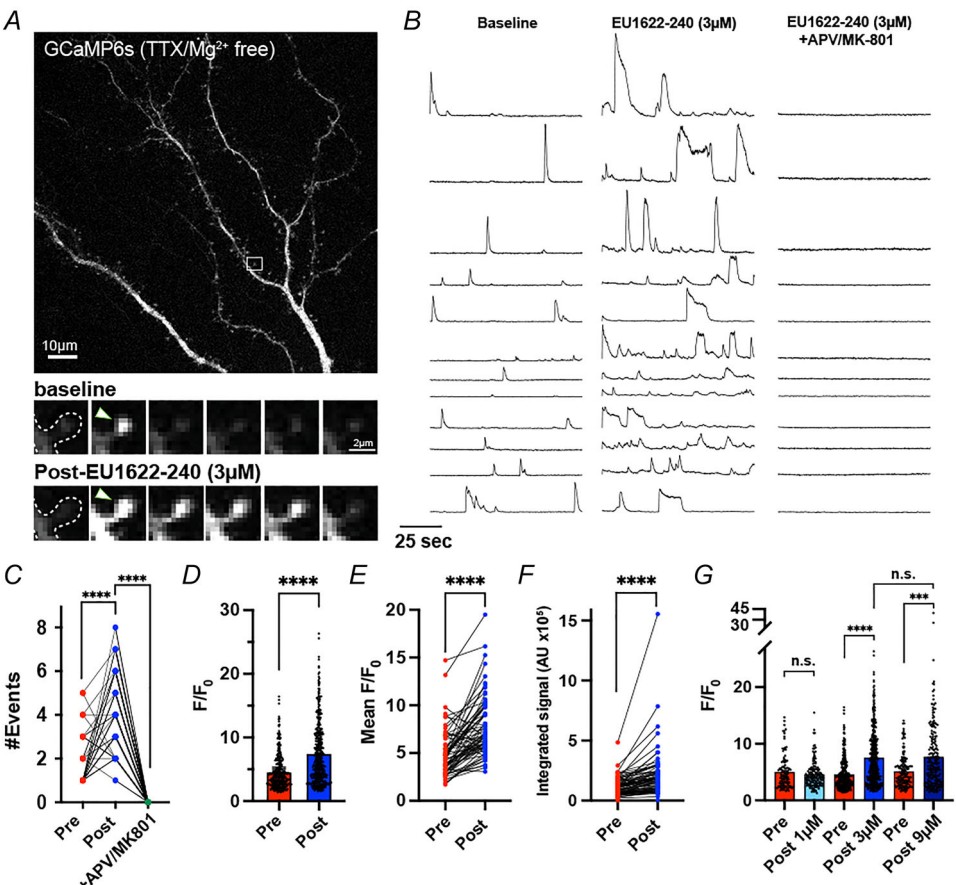

**Figure 7. EU1622-240 increases NMDAR-mediated Ca$^{2+}$ entry in response to quantal synaptic glutamate release**

*A*, a dissociated hippocampal neuron transfected with the genetically encoded Ca$^{2+}$ indicator GCaMP6s is shown. The time series below compares quantal Ca$^{2+}$ transients (white arrowheads) at the same dendritic spine (white box in image above), before (top row) and ~8 min after application of 3 μM EU1622-240 (bottom row). Dashed line denotes the outline of the cell, drawn based on a co-expressed mCherry cell fill. Time between frames is 3.5 s to demonstrate prolonged responses in the presence of EU1622-240. Scale bars: 2 μm in time series; 10 μm in full field image. *B*, traces show paired GCaMP6s intensity (in arbitrary units) at the same 12 dendritic spines before (left traces), after application of 3 μM EU1622-240 (middle traces) and after 100 μM D,L-APV and 20 μM MK-801 in the continued presence of 3 μM EU1622-240. *C*, quantification of the total number of events during 1.8 min of imaging (750 frames at 7 Hz) at the same dendritic spines before (Pre, red), after application of 3 μM EU1622-240 (Post, blue) and after addition of 100 μM D,L-APV and 20 μM MK-801 in the continued presence of 3 μM EU1622-240 (green). Thirty-nine spines from four neurons were analysed. ****$P < 0001$, Wilcoxon matched-pairs signed rank test. *D*, quantification of peak Ca$^{2+}$ transient amplitudes from all events (expressed as peak fluorescence divided by baseline fluorescence, $F/F_0$) recorded before (Pre, red) and after application of 3 μM EU1622-240 (Post, blue). Seventy-two synapses were analysed from five neurons. ****$P < 0.0001$, two-tailed Mann–Whitney *U* test. *E*, same as *C*, except showing a paired comparison of the mean peak Ca$^{2+}$ amplitudes recorded from the same synapses before (Pre, red) and after addition of 3 μM EU1622-240 (Post, blue). ****$P < 0.0001$, Wilcoxon matched-pairs signed rank test. *F*, paired comparison of the summated GCaMP6s Ca$^{2+}$ responses over the 1.8 min imaging session from the same individual spines before (Pre, red) and after application of 3 μM EU1622-240 (Post, blue). ****$P < 0.0001$, Wilcoxon matched-pairs signed rank test. *G*, peak Ca$^{2+}$ transient response amplitudes before and after application of 1, 3 and 9 μM EU1622-240. Data for the 3 μM dose is repeated from *C*, for comparison. For 1 μM: 25 synapses from three neurons, and n.s. indicates not significant; for 9 μM: 27 synapses from three neurons, ***$P = 0.0004$, two-tailed Mann–Whitney *U* test.

neurons should increase the frequency of IPSCs in CA1 pyramidal cells. To evaluate this hypothesis, sIPSC recordings were performed on CA1 pyramidal cells at a holding potential of +10 mV under voltage clamp. Figure 9*A1* shows representative traces before (upper trace) and during application of 3 μM EU1622-240 (lower trace). Figure 9*A2* shows that EU1622-240 shifted the sIPSC inter-event interval significantly to the left

in comparison to baseline ($P < 0.0001$, two-sample Kolmogorov–Smirnov test).

EU1622-240 significantly increased IPSC frequency to $1.8 \pm 0.56$-fold of control ($n = 8$; Fig. 9*B* and *C*; Student's unpaired $t$ test, $P = 0.000169 < 0.025$). In contrast, there was no effect of EU1622-240 on IPSC frequency when D,L-APV was present to block NMDARs (treatment/control was $0.76 \pm 0.48$-fold, $n = 14$, $P = 0.34$).

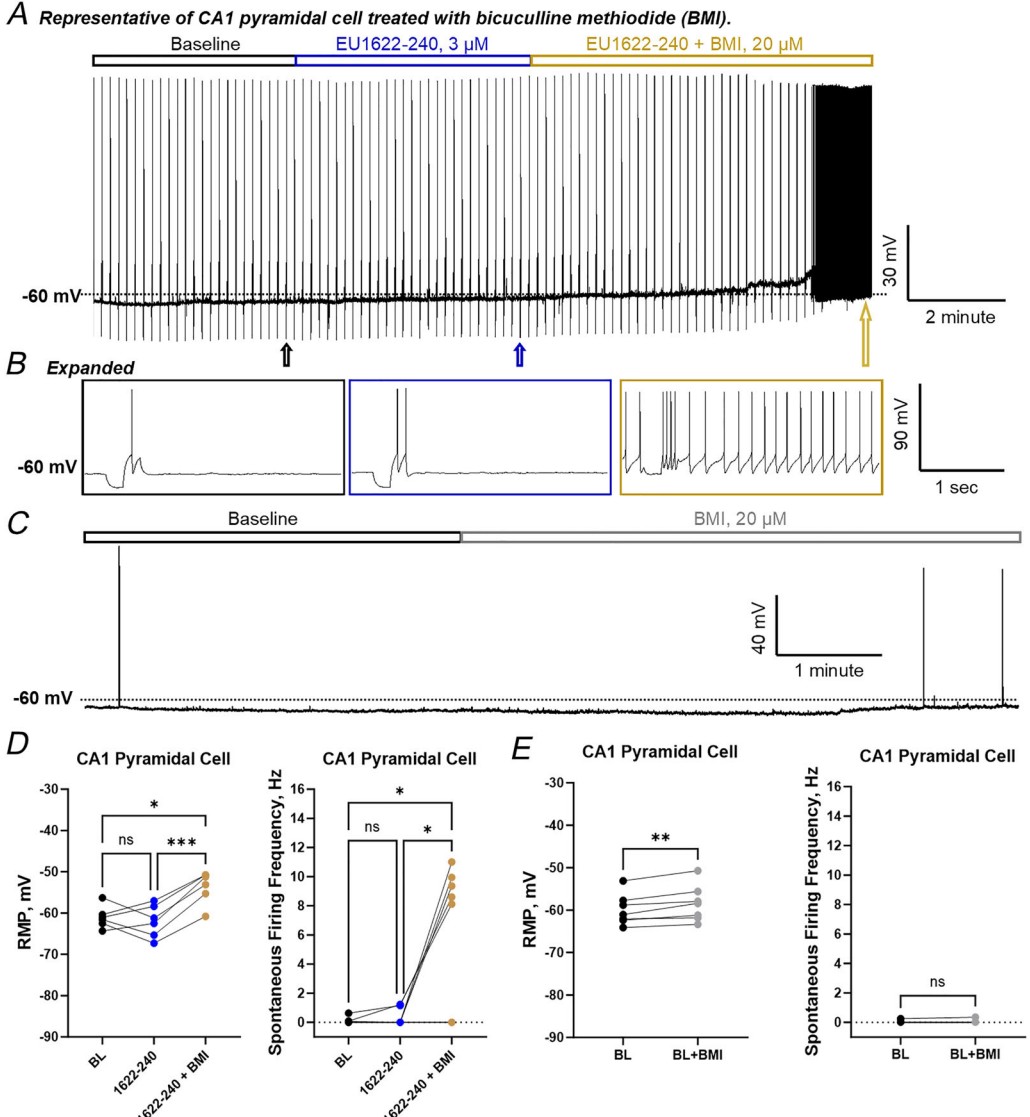

**Figure 8. Bicuculline methiodide alters the EU1622-240 response of CA1 pyramidal cells**
*A*, a recording of membrane potential under current clamp is shown for baseline (black outlined bar), 3 μM EU1622-240 (blue outlined bar) and co-application of 3 μM EU1622-240 with 20 μM bicuculline methiodide (BMI; gold outlined bar). *B*, expanded recordings from the end of each treatment period are shown; short hyperpolarizing current injections were applied intermittently throughout the recording period. *C*, a representative current-clamp recording is shown for a CA1 pyramidal cell for a baseline period (black outlined bar) followed by wash-in of 20 μM BMI (grey outlined bar). *D*, summary of the actions of EU1622-240 and EU1622-240 + BMI on resting membrane potential and spontaneous firing frequency from CA1 pyramidal cells is shown; $n = 6$ cells; *$P \leq 0.05$ and ***$P \leq 0.001$, repeated-measures one-way ANOVA with Tukey's multiple comparisons test. *E*, a summary of the effects of BMI alone on CA1 pyramidal cell resting membrane potential and spontaneous firing frequency is shown; $n = 7$ cells; **$P \leq 0.01$; ns, no significance by Student's paired $t$ test.

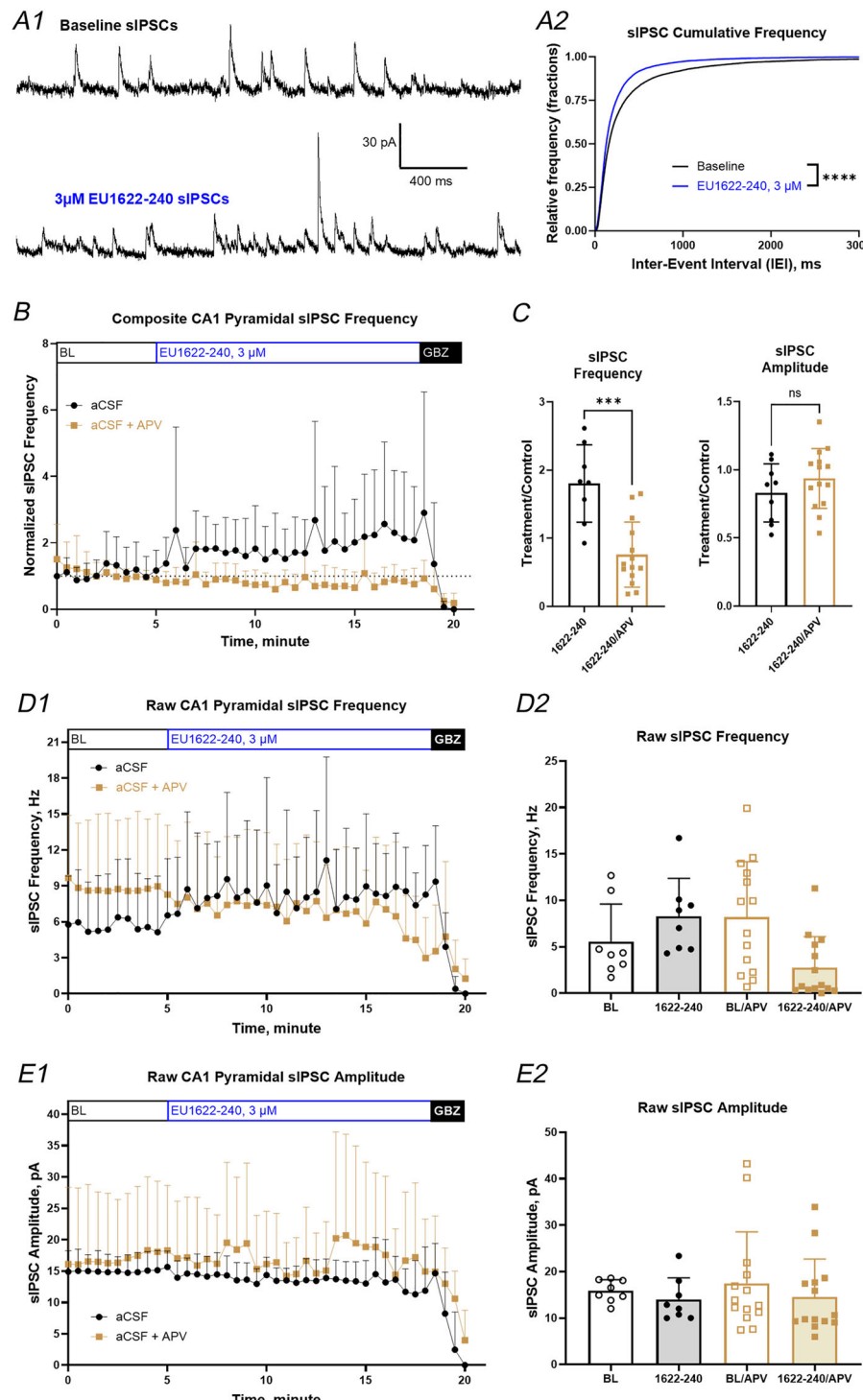

**Figure 9. Spontaneous IPSC frequency recorded in CA1 pyramidal cells was increased by EU1622-240**
*A*, representative sIPSC recordings are shown from CA1 pyramidal cells before (*A1*, upper trace) and during the application of 3 μM EU1622-240 (*A1*, lower trace); the cumulative sIPSC inter-event interval (IEI) is plotted in *A2*). *B*, the normalized sIPSC instantaneous frequency either with (gold squares) or without 400 μM D,L-APV (black circles) added to the aCSF. Open bars indicate the baseline period (BL) and the wash-in of 3 μM EU1622-240 (blue); the solid black bar indicates the wash-in of 10 μM gabazine (GBZ) together with 3 μM EU1622-240. The last 1 min of each treatment was used for analysis. *C*, the average fold-changes over baseline response in sIPSC frequency (left) and amplitude (right) of spontaneous IPSCs are shown for recordings in the absence or presence of D,L-APV. *D* and *E*, the raw sIPSC frequency (*D1* and *D2*) and amplitude (*E1* and *E2*) are shown. ***$P \leq 0.001$ by Student's unpaired *t* test of fold-change over baseline for test conditions. $n = 8$ for aCSF and 14 for aCSF + D,L-APV, mean ± SD.

**Table 4. Bicuculline methiodide alters the effects of EU1622-240 on CA1 pyramidal cell firing activity**

| Conditions | Pyramidal cell | | |
|---|---|---|---|
| | RMP (mV) | Spontaneous firing frequency (Hz) | *n* |
| Baseline A | −52 ± 2.7 | 0.10 ± 0.2 | 6 |
| 3 μM EU1622-240[a] | −53 ± 3.9 | 0.27 ± 0.6 | 6 |
| EU1622-240 + BMI[a] | −46 ± 3.9 (0.0213) | 5.2 ± 4.9 (*0.0292*) | 6 |
| Baseline B | −52 ± 3.7 | 0.031 ± 0.1 | 7 |
| Baseline + BMI[b] | −51 ± 4.3 (0.0057) | 0.043 ± 0.1 | 7 |

Baseline A and Baseline B are from two independent experiments. Abbreviation: BMI, bicuculline methiodide; RMP, resting membrane potential; μM: micromolar; Hz: hertz; mV: millivolts.
[a] Data are presented as the mean ± SD (*P*-value), where the *P*-value was calculated by comparing EU1622-240 treatment and EU1622-240 + BMI with baseline A; repeated-measures ANOVA with Tukey's multiple comparisons test.
[b] Data are presented as mean ± SD (*P*-value), where the *P*-value was calculated by comparing BMI treatment with baseline B by Student's paired *t* test.

There was no detectable difference in sIPSC amplitude in EU1622-240 in the absence (0.91 ± 0.21-fold of baseline, Student's paired *t* test, $P = 0.086$) or in the presence of D,L-APV (1.0 ± 0.22-fold of baseline, $P = 0.33$). When parallel experiments were performed with 400 μM D,L-APV present in the aCSF, D,L-APV itself did not detectably alter mean sIPSC frequency (control 8.3 ± 4.1 Hz, D,L-APV 9.0 ± 6.0 Hz; Student's paired *t* test, $P = 0.307$; Fig. 9*B*). The un-normalized sIPSC amplitude and frequency are shown in Fig. 9*D* and *E*. These data suggest that tonically active NMDARs do not impact spontaneous IPSC frequency in resting conditions. However, EU1622-240 can increase spontaneous IPSC frequency in an NMDAR-dependent manner, consistent with EU1622-240 enhancement of NMDAR function on interneurons.

If EU1622-240 selectively enhances inhibitory signalling onto CA1 pyramidal cells, we should be able to detect a shift in the excitatory–inhibitory balance in CA1 pyramidal cells. To test this idea, we recorded EPSPs and IPSPs from CA1 pyramidal cells in response to Schaffer collateral afferent stimulation. This should produce a compound postsynaptic potential that consists of a glutamate-mediated EPSP followed by a GABA-mediated IPSP. We used the amplitude of this evoked EPSP and IPSP to assess the excitatory–inhibitory balance in CA1 pyramidal cells before and during 3 μM EU1622-240 treatment. Two types of stimulation positions were used: stimulating at the CA3 region (distal stimulus) and at the CA1 region (local stimulus). Figure 10*A* shows representative recordings from the distal stimulus with baseline recording before EU1622-240 application (black), and after 10 min of EU1622-240 treatment. We observed a significant reduction of evoked EPSP amplitude (baseline 2.1 ± 1.3 mV, EU1622-240 1.3 ± 0.88 mV,

$P = 0.0044 < 0.025$), an increase in the evoked IPSP amplitude (baseline 0.93 ± 0.71 mV, EU1622-240 2.4 ± 1.8 mV, $P = 0.0037$), hence a significantly decreased EPSP/IPSP ratio (baseline 3.7 ± 3.4-fold, EU1622-240 0.77 ± 0.91-fold, $P = 0.0075 < 0.0125$, $n = 10$, Student's paired *t* test with Bonferroni correction; Fig. 10*B–D*). Figure 10*E* shows representative recordings from a local stimulus with baseline recording before EU1622-240 application (black) and after 10 min of EU1622-240 treatment. Interestingly, in contrast to distal stimulation, we observed a significant increase of evoked EPSP amplitude (baseline 5.29 ± 1.99 mV, EU1622-240 7.45 ± 2.15 mV, $P = 0.00041$), an increase in the evoked IPSP (baseline 2.0 ± 0.41 mV, EU1622-240 3.0 ± 0.53 mV, $P = 0.0038 < 0.025$), with no significant change of EPSP/IPSP ratio for local stimulation (baseline 2.8 ± 1.4-fold, EU1622-240 2.5 ± 0.53-fold, $P = 0.410$, $n = 8$, Student's paired *t* test with Bonferroni correction; Fig. 10*F–H*). These data are consistent with enhanced recruitment of feedforward inhibition for distal stimulation, whereas distal excitatory drive is diminished in comparison to proximal stimulation in acutely prepared slices (Dingledine, Roth et al. 1987).

### EU1622-240 potentiates field EPSPs

We also investigated the effect of 3 μM EU1622-240 on the local evoked field potentials in mouse hippocampal slices. Given the ability of EU1622-240 to enhance NMDAR function, we tested whether this modulator could influence cellular models of plasticity, such as LTP. We tested EU1622-240 on brain slices produced from WT mice. We measured the slope of the evoked fEPSP recorded in the CA1 stratum radiatum to determine whether EU1622-240-induced potentiation of NMDAR can alter the fEPSP. Figure 11 shows the effect of

EU1622-240 on the slope of the fEPSP produced in the CA1 region with electric stimulation of Schaffer collateral fibres. We recorded evoked fEPSPs from slices treated with either vehicle (0.3% DMSO, labelled as baseline) or 3 μM EU1622-240 applied for 60 min. Figure 11*A* shows the slope of the field response with 60 min baseline (last 10 min shown, black trace, BL: aCSF with 0.3% DMSO) followed by 60 min of 3 μM EU1622-240 application (blue trace, EU240: EU1622-240). Figure 11*B* shows full field potential response expanded traces for 10 min baseline (black trace, aCSF with 0.3% DMSO) followed by 20 and 30 min of 3 μM EU1622-240 application (coloured traces). The baseline fEPSP slope was $1.3 \pm 1.3$ mV/ms. Following the wash-in of 3 μM EU1622-240, the fEPSP slope was (in millivolts per millisecond): $1.5 \pm 1.4$ (10 min), $1.9 \pm 1.8$ (20 min), $2.4 \pm 2.1$ (30 min), $2.6 \pm 2.4$ (40 min), $2.7 \pm 2.4$ (50 min), $2.7 \pm 2.7$ (60 min) and $3.7 \pm 3.0$ (65 min) after EU1622-240 administration. At 3 μM, EU1622-240 significantly increased the fold-change of field amplitude (repeated-measures one-way ANOVA with Dunnett's multiple comparison test [$F(1.178, 8.247) = 6.653$, $P = 0.0285$; Fig. 11*C*]. Representative currents and data points were collected by averaging the last 2 min of each treatment period.

Given the ability of EU1622-240 to produce a long-lasting increase in fEPSP slope, we predicted that it would occlude LTP if it engaged similar mechanisms to LTP. We therefore measured the effect of 3 μM EU1622-240 on LTP induced by four sets of theta-burst stimulation. We observed that 3 μM 1622-240 disrupted potentiation of the slope of fEPSP in response to theta-burst stimulation, when compared with the fEPSP slope on the fourth theta burst at 100 min administration (Fig. 11*D* and *E*; $P = 0.0992$, Student's unpaired *t* test).

## Discussion

The most important finding from this study is that the effects of EU1622-240 NMDAR potentiation depend on the type of neuron, with preferential enhancement of the excitability of hippocampal interneurons over principal cells. We found that EU1622-240 depolarized CA1 interneuron RMP and increased firing frequency, without changing either parameter for CA1 pyramidal cells. This result aligns with the properties of this PAM determined *in vitro*, which possesses preferential enhancement of GluN2D-containing NMDARs that are expressed in interneurons (Dubois & Liu 2021; Dubois, Lachamp et al. 2016; Gawande, KK et al. 2023; Hanson, Armbruster

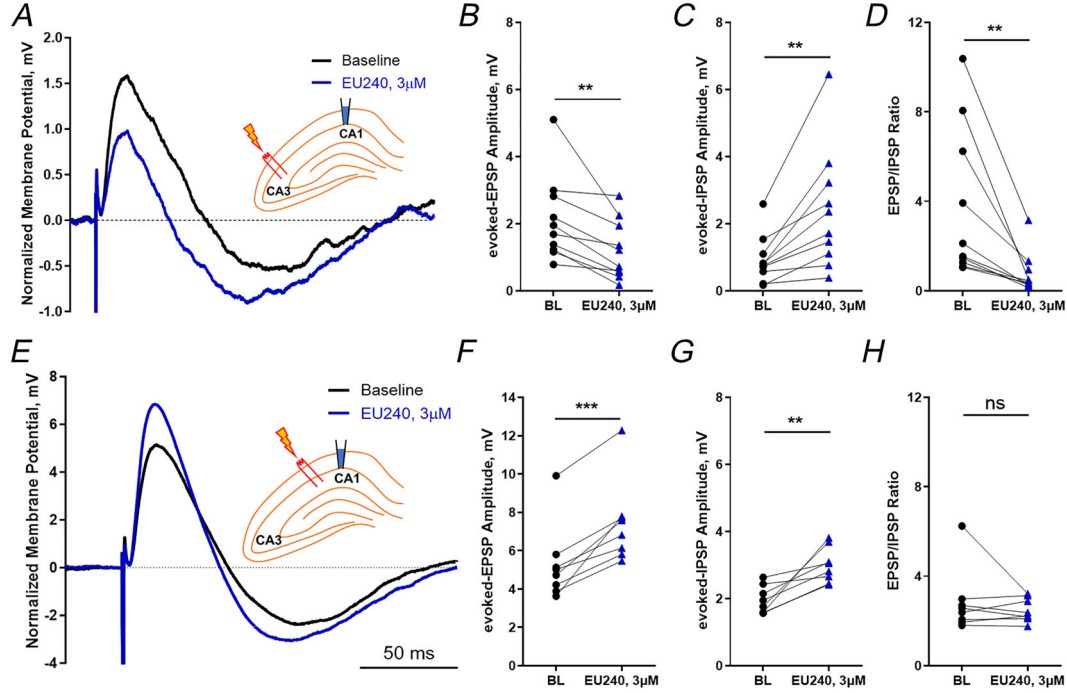

**Figure 10. EU1622-240 alters the EPSP/IPSP ratio from CA1 pyramidal cells**
*A* and *E*, representative voltage recordings of an evoked EPSP/IPSP from a CA1 pyramidal cell in response to distal (upper panel, *A–D*) or proximal (lower panel, *E–H*) stimulation. Baseline response (BL; black) and 3 μM EU1622-240-treated response (EU240; blue) after 10 min administration are shown. The evoked EPSP amplitude (*B* and *F*), evoked IPSP amplitude (*C* and *G*), and EPSP/IPSP ratio (*D* and *H*) are plotted. **$P \leq 0.01$, Student's paired *t* test; *n* = 10 cells. Recordings were made in current-clamp mode with current injection to bring the membrane potential at rest to −60 mV. The stimulus strength was adjusted such that the initial PSP depolarization was ∼2 mV.

et al. 2019; Mellone, Zianni et al. 2019; Perszyk, DiRaddo et al. 2016) over NMDARs in principal cells, which contain GluN2A and GluN2B (Ladagu, Olopade et al. 2023; Shipton & Paulsen 2014). In addition, there was potentiation by EU1622-240 of EPSC amplitude and charge transfer in both interneurons and CA1 pyramidal cells. We also detected an increase in Ca$^{2+}$ signalling through miniature NMDAR-mediated EPSCs in cultured neurons, consistent with some meaningful enhancement of excitatory drive onto pyramidal cells. This is likely to reflect the actions of EU1622-240 at GluN2B-containing receptors. However, neither pyramidal cells nor interneurons showed a detectable change in the response time course, as measured by the weighted $\tau$ describing the exponential decay of the NMDAR-mediated EPSC.

The result of enhanced interneuron excitability could be observed as an increase in spontaneous IPSCs onto CA1 pyramidal cells in hippocampal slices. An important implication of this finding is that NMDARs can influence interneuron excitability and, consequently, inhibitory drive onto CA1 pyramidal cells. The basis for this enhanced excitatory drive could reflect at least two processes. It could arise from direct enhancement of the NMDAR-mediated component of excitatory synaptic transmission, potentially owing to the contribution of GluN2D-containing NMDARs to excitatory synaptic transmission onto interneurons. This could take the form of GluN2D-containing pre- or postsynaptic NMDARs. In addition, the depolarization and decrease in input resistance are consistent with extracellular glutamate producing modest tonic activation of extrasynaptic EU1622-240-bound GluN2D-containing NMDARs, which are more sensitive to glutamate (Erreger, Geballe et al. 2007), allowing them to be activated by lower concentrations of agonists (Ullman, Perszyk et al. 2024). Removal of the need for co-agonist binding for NMDAR activation by EU1622-240 might also contribute to this effect (Ullman, Perszyk et al. 2024). In addition, a higher ambient glutamate microenvironment on inhibitory interneurons than on excitatory pyramidal cells could contribute to this preferential enhancement of interneuron firing by EU1622-240 (Yao, Grand et al. 2018).

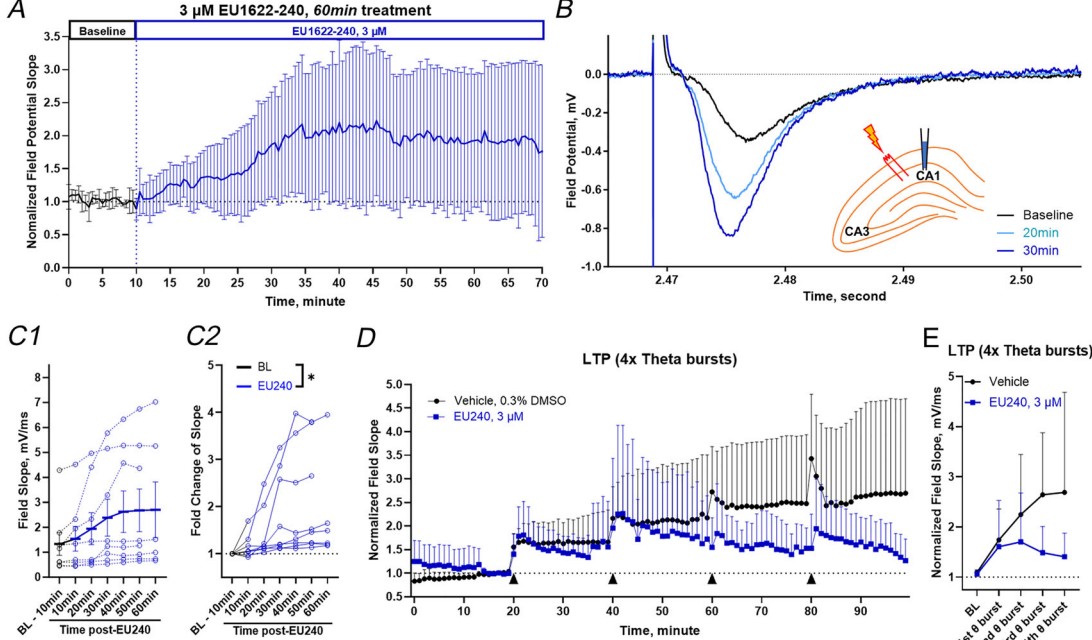

**Figure 11. Extracellular field potential recordings of EPSPs in response to 3 μM EU1622-240**
*A*, the slope of field EPSPs normalized to baseline is shown during a 10 min baseline followed by 60 min application of 3 μM EU1622-240. *B*, representative recordings of evoked field EPSPs are shown at the indicated times from *A*. *C*, plots are shown of the raw field EPSP slope *C1*) and fold-change of field response slope (*C2*) for each recorded slice. Responses shown and data were collected by averaging the last 5 min of baseline for 10 min interval analysis periods (*n* = 8). At 3 μM, EU1622-240 significantly increased the fold-change of field EPSP amplitude; repeated-measures one-way ANOVA with Dunnett's multiple comparison test [*F*(1.178, 8.247) = 6.653, *P* = 0.0285]. *D*, a plot of the normalized field response slope of LTP recording with four theta-burst stimulations; *n* = 8 vehicle (black, 0.3% DMSO) and *n* = 7 (EU1622-240 treatment, blue, 3 μM). Black triangles indicate the time of each theta-burst stimulation. *E*, the normalized field response slope of LTP is shown by analysing the last 3 min after the fourth theta burst (*P* = 0.0992, Student's unpaired *t* test). All tested slices in *D* had been placed in the treatment solution (EU1622-240, vehicle) for 20 min prior to the start of the 20 min baseline recording period. Data are presented as mean ± SD. Abbreviation: BL, baseline (aCSF with 0.3% DMSO).

The preferential enhancement of interneuron excitability by EU1622-240 could reflect the higher firing rate of interneurons with a more depolarized RMP, which could reduce $Mg^{2+}$ blockade of NMDARs located on these interneurons, allowing for a higher level of basal NMDAR synaptic currents on interneurons (Cohen, Tsien et al. 2015). Another NMDAR PAM, GNE-0723 (Villemure, Volgraf et al. 2017; Volgraf, Sellers et al. 2016), has shown a similar effect that predominantly enhances CA1 interneurons over pyramidal cells. Moreover, a similar preferential effect has been observed with another NMDAR positive modulator, PTC-174, which favours potentiation of GluN2D-containing receptors (Yi, Rouzbeh et al. 2020). Activation of synaptic and extrasynaptic NMDARs can trigger receptor internalization (Nong, Huang et al. 2003), and it is possible that CA1 pyramidal cells and interneurons might have different degrees of NMDAR internalization followed by the receptor activation, which could affect cell excitability differently after the 10 min EU1622-240 application in our experiment. Although GluN2A- and GluN2B-containing NMDARs on CA1 pyramidal cells also should show increased sensitivity, this does not appear to produce a detectable tonic depolarization or change in input resistance. We also do not see increases in resting $Ca^{2+}$ for EU1622-240 potentiation of NMDAR function in response to spontaneous synaptic glutamate release during $Ca^{2+}$ real-time imaging in cultured neurons. One potential reason for the lack of discernible effects on CA1 pyramidal cell excitability by EU1622-240, which can enhance efficacy at GluN2B-containing receptors and slow NMDAR deactivation following glutamate removal for both GluN2A and GluN2B-containing NMDARs, could be the enhanced inhibitory drive onto the CA1 pyramidal cell by interneurons. We tested this idea by repeating experiments in the presence of the GABA$_A$ receptor inhibitor bicuculline. In these conditions, EU1622-240 was able to depolarize CA1 pyramidal cells and produce a significant increase in spike firing. However, the interpretation is confounded by the actions of bicuculline alone, which depolarize neurons and thus will increase NMDAR-mediated inward currents through diminished $Mg^{2+}$ block.

GluN2C-containing NMDARs are expressed on astrocytes within the CA1 region (Chipman, Fung et al. 2021). Given that EU1622-240 potentiates GluN2C-containing NMDARs ($270 \pm 27\%$ compared with control) (Fritzemeier, Akins et al. 2025; Ullman, Perszyk et al. 2024), the positive modulation of GluN2C-containing astrocytic NMDARs could contribute in some way to the actions we observed on CA1 pyramidal cells and CA1 interneuron firing rates and membrane potential. Data reported by Fritzemeier, Akins et al. (2025) indicate that EU1622-240 does not show significant off-target action on 66 ion channels/G-protein-coupled receptors/transporters that have been screened (Fritzmeier et al., 2025). We also showed that *GluN2D* knockout mice abolish the actions of EU1622-240, arguing against a prominent role for potentiation of astrocytic GluN2C-containing NMDARs by EU1622-240.

Our EPSP/IPSP experiment showed an interesting effect on the evoked fEPSP. When stimulating Schaffer collateral fibres distal in the CA3 region, 3 μM EU1622-240 decreased the evoked EPSP while increasing the evoked IPSP. In contrast, when local stimulation was applied close to the CA1 pyramidal cell being recorded, 3 μM EU1622-240 increased both the evoked EPSP and the evoked IPSP. We conclude that this is attributable to the enhancement of the feedforward inhibition network in both situations by 3 μM EU1622-240, whereas we see only an enhanced EPSP for proximal stimulation that produces robust EPSPs (Dingledine, Roth et al. 1987). This might also be impacted, in part, by the mechanisms underlying the EU1622-240-induced LTP. Furthermore, with distal stimulation, orthodromic evoked excitatory signals travel along the Schaffer collateral fibres, which excite CA1 pyramidal cells together with a number of CA1 GABAergic interneurons. Owing to the EU1622-240 potentiation of GluN2D, which are expressed on CA1 interneurons, the interneurons will be potentiated effectively. A feedforward inhibition creates a monosynaptic IPSP onto CA1 pyramidal cells. Such monosynaptic IPSPs can superimpose on and thus diminish the amplitude of the EPSP component and increase the IPSP component of the compound postsynaptic potential. Given that EU1622-240 also potentiates GluN2B, which is expressed on CA1 pyramidal cells, we did see an increased EPSP during EU1622-240 administration.

## Conclusion

This project focused on the effects of an NMDAR positive allosteric modulator, EU1622-240, on hippocampal neurons within the postnatal day 17–22 developmental window because this time period represents a foundational period in circuit formation that is essential for lifelong learning, memory and cognitive function. These data will thus help us to understand the role of EU1622-240 potentially in targeting both early-onset neuropathological conditions (schizophrenia and epilepsy). Although these results might also be relevant for later-onset disorders that also involve perturbation of NMDAR function (Alzheimer's disease and age-related cognitive decline), additional studies will be required to evaluate potential actions of this class of NMDAR PAMs on conditions involving older ages.

The novel class of NMDAR PAM exemplified by the tool compounds EU1622-240 shows a preferential

potentiating effect onto GluN2D-containing CA1 interneurons over GluN2A/GluN2B-expressing CA1 pyramidal cells, which could have useful actions in modulating circuit function in a variety of pathological conditions. There were some actions at excitatory synapses, however, and EU1622-240 had an ability to induce a form of enhanced potentiation that occluded LTP. This suggests that this series of modulators might have therapeutically relevant effects on NMDARs that mitigate clinical problems, including anxiety, treatment-resistant depression or cognitive dysfunction (Hanson, Yuan et al. 2024). Furthermore, enhancing GABA signalling can effectively reduce seizure-like activity in the mouse hippocampus (Liddiard, Suryavanshi et al. 2024). The dysfunction in the excitation–inhibition balance is apparent in rodent models of schizophrenia (Gawande, KK et al. 2023; Lisman, Coyle et al. 2008), with a reduced inhibitory signalling level. Thus, PAMs such as EU1622-240 might be effective, via their enhancement of inhibitory tone, for disorders involving diminished circuit inhibition.

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

## Additional information

### Data availability statement

All data supporting the results of this manuscript are presented within the paper itself. Source data for all figures within this manuscript are available from the corresponding author upon reasonable request.

### Competing interests

S.F.T. is a member of the SAB for Eumentis Therapeutics, Neurocrine Biosciences, a member of the MAB for the GRIN2B Foundation and the CureGRIN Foundation, a consultant for GRIN Therapeutics, Seyltx, a co-founder of NeurOp Inc. and AgriThera, a member of the Board of Directors for NeurOp Inc., a Principal Investigator on a grant from GRIN Therapeutics, and co-inventor on Emory-owned Intellectual Property that includes positive allosteric modulators of NMDAR function. H.Y. is Principal Investigator on research grants from Sage Therapeutics and GRIN Therapeutics to Emory. D.C.L. is a member of the Board of Directors for NeurOp Inc., co-inventor on Emory-owned Intellectual Property that includes positive allosteric modulators of NMDAR function. R.G.F., N.S.A. and S.P. are co-inventors on Emory-owned Intellectual Property that includes positive allosteric modulators of NMDAR function.

### Author contributions

H.X. designed the work, acquired, analysed and interpreted data for the work, and wrote the manuscript. T.G.B., B.R.B., O.P. and M.J.K. designed the work, acquired, analysed and interpreted data for the work, and wrote the manuscript. L.Z. and K.Y. designed the work, acquired, analysed and interpreted data for the work. C.R.C. designed the work, interpreted data for the work, and wrote the manuscript. R.G.F., N.S.A., S.P., P.J.A., and D.C.L. synthesized EU1622-240, and wrote the manuscript. H.Y. and S.F.T. designed the work, analysed and interpreted data for the work, and wrote the manuscript. All authors approved the final version of the manuscript and agree to be accountable for all aspects of the work in ensuring that questions related to the accuracy or integrity of any part of the work are appropriately investigated and resolved. All persons designated as authors qualify for authorship, and all those who qualify for authorship are listed.

### Funding

This work was supported by the NIH-NINDS (NS111619 SFT), NICHD (HD082373 HY), NIH-NIMH (H134957 MJK), NIH-NINDS (NS116879 MJK), and research grants to Emory University from Janssen Research and Development LLC and

Biogen (Principal Investigator S.F.T.). The content of this publication does not necessarily reflect the views or policies of the Department of Health and Human Services, nor does the mention of trade names, commercial products or organizations imply endorsement by the US Government.

## Acknowledgements

We thank Jing Zhang, Sukhan Kim, Christina Winborn and Eva Sarai Diaz for excellent technical assistance.

## Author's present address

Lu Zhang: Department of Neurology, National Centre for Neurological Disorders, Huashan Hospital, Shanghai Medical College, Fudan University, Shanghai 200040, China.

## Keywords

GluN2C, GluN2D, hippocampus, interneuron, NMDA receptor, positive allosteric modulator, pyramidal cell

## Supporting information

Additional supporting information can be found online in the Supporting Information section at the end of the HTML view of the article. Supporting information files available:

**Peer Review History**
**Supporting Video S1**
**Supporting Video S2**

