## [Peer Review History · The Journal of Physiology]

Enhancement of hippocampal interneuron excitability by NMDA receptor positive allosteric modulation

Hao Xing, Tue G. Banke, Lu Zhang, Kuai Yu, Chad R. Camp, Russell G Fritzemeier, Nicholas Akins, Srinu Paladugu, Paul J Arcoria, Brian R Brady, Olga Prikhodko, Matthew Kennedy, Dennis C. Liotta, HONGJIE YUAN, and Stephen F. Traynelis
DOI: 10.1113/JP289774

Corresponding author(s): Stephen Traynelis (strayne@emory.edu)

The following individual(s) involved in review of this submission have agreed to reveal their identity: Zhuo Huang (Referee #1); Lonnie P Wollmuth (Referee #2)

Review Timeline:

Submission Date:	25-Jul-2025
Editorial Decision:	15-Aug-2025
Revision Received:	03-Sep-2025
Accepted:	15-Sep-2025

Senior Editor: Jing-Ning Zhu

Reviewing Editor: Ming Yi

Transaction Report:

Dear Dr Traynelis,

Re: JP-RP-2025-289774 "**Enhancement of hippocampal interneuron excitability by NMDA receptor positive allosteric modulation**" by Hao Xing, Tue G. Banke, Lu Zhang, Kuai Yu, Chad R. Camp, Russell G Fritzemeier, Nicholas Akins, Srinu Paladugu, Paul J Arcoria, Brian R Brady, Olga Prikhodko, Matthew Kennedy, Dennis C. Liotta, HONGJIE YUAN, and Stephen F. Traynelis

Thank you for submitting your manuscript to The Journal of Physiology. It has been assessed by a Reviewing Editor and by 2 expert referees and we are pleased to tell you that it is potentially acceptable for publication following satisfactory major revision.

REVISION CHECKLIST:

We look forward to receiving your revised submission.

Yours sincerely,

Jing-Ning Zhu
Senior Editor
The Journal of Physiology

REQUIRED ITEMS

- Author photo and profile. First or joint first authors are asked to provide a short biography (no more than 100 words for one author or 150 words in total for joint first authors) and a portrait photograph. These should be uploaded and clearly labelled together in a Word document with the revised version of the manuscript. See Information for Authors for further details.

- You must start the Methods section with a paragraph headed Ethical approval (https://jp.msubmit.net/cgi-bin/main.plex?form_type=display_requirements#methods).

Research must comply with The Journal's policies regarding animal experiments (<https://physoc.onlinelibrary.wiley.com/hub/animal-experiments>) and adherence to these policies must be stated in the manuscript.

Authors should confirm in their Methods section that their experiments were carried out according to the guidelines laid down by their institution's animal welfare committee, including an ethics approval reference number. The Methods section must contain a statement about access to food, water and housing, details of the anaesthetic regime: anaesthetic used, dose and route of administration, and method of killing the experimental animals.

- Please upload separate high-quality figure files via the submission form.

- Please ensure that any tables are editable and in Word format, and wherever possible, embedded in the article file itself.

- Please ensure that the Article File you upload is a Word file.

- A Data Availability Statement is required for all papers reporting original data. This must be in the Additional Information section of the manuscript itself. It must have the paragraph heading 'Data Availability Statement'. All data supporting the results in the paper must be either: in the paper itself; uploaded as Supporting Information for Online Publication; or archived in an appropriate public repository. The statement needs to describe the availability or the absence of shared data. Authors must include in their statement: a link to the repository they have used, or a statement that it is available as Supporting Information; reference the data in the appropriate sections(s) of their manuscript; and cite the data they have shared in the References section. Whenever possible, the scripts and other artefacts used to generate the analyses presented in the paper should also be publicly archived. If sharing data compromises ethical standards or legal requirements then authors are not expected to share it, but must note this in their statement. For more information, see our Statistics Policy.

- Please include an Abstract Figure file, as well as the Figure Legend text within the main article file. The Abstract Figure is a piece of artwork designed to give readers an immediate understanding of the research and should summarise the main conclusions. If possible, the image should be easily 'readable' from left to right or top to bottom. It should show the physiological relevance of the manuscript so readers can assess the importance and content of its findings. Abstract Figures should not merely recapitulate other figures in the manuscript. Please try to keep the diagram as simple as possible and without superfluous information that may distract from the main conclusion(s). Abstract Figures must be provided by authors no later than the revised manuscript stage and should be uploaded as a separate file during online submission labelled as File Type 'Abstract Figure'. Please also ensure that you include the figure legend in the main article file. All Abstract Figures should be created using BioRender. Authors should use The Journal's premium BioRender account to export high-resolution images. Details on how to use and access the premium account are included as part of this email.

EDITOR COMMENTS

Reviewing Editor:

Overall both reviewers consider that this manuscript is of high impact to the field and have provided comments for authors to revise the manuscript. Please carefully read the reviews, revise the manuscript (including additional experiments if necessary) and submit a revised manuscript together with a point-to-point response letter.

Please also see 'Required Items' above.

Senior Editor:

Thank you for submitting your manuscript to The Journal of Physiology. Your study on the selective modulation of GluN2D-containing NMDARs by EU1622-240 in hippocampal microcircuits has been evaluated by two expert reviewers and a reviewing editor. Both reviewers and the editor found your work to be of high quality and potential interest to the field, but they have raised several substantive concerns that need to be addressed before the manuscript can be considered for acceptance. Please submit a revised manuscript with a point-by-point response to the reviewers' comments. Please consider clarifying the relative contributions of GluN2B and GluN2D to EU1622-240's effects (on Ca²⁺ entry) by including experiments in GluN2B-KO mice or immunostaining for these subunits.

REFEREE COMMENTS

Referee #1:

Xing and colleagues' research significantly advances our understanding of NMDA receptor (NMDAR) modulation in hippocampal microcircuits. Their study demonstrates that the positive allosteric modulator EU1622-240 selectively enhances the excitability of CA1 stratum radiatum interneurons while leaving CA1 pyramidal cells largely unaffected. Using whole-cell patch-clamp recordings, calcium imaging, and field-potential assays in mouse hippocampal slices and cultured neurons, they show that EU1622-240 potentiates GluN2D-containing NMDARs expressed predominantly on interneurons, leading to membrane depolarization, increased spike firing, and elevated inhibitory output onto pyramidal cells. This shift in the excitation-inhibition balance is accompanied by a marked rise in spontaneous inhibitory postsynaptic currents (sIPSCs) recorded from CA1 pyramidal cells. In addition, the compound induces a long-lasting potentiation of synaptic transmission that occludes conventional theta-burst LTP, suggesting a novel form of NMDAR-dependent plasticity. Critically, these effects are abolished in GluN2D-knockout mice, confirming the subunit specificity of EU1622-240. Overall, this work provides compelling evidence that selective potentiation of GABAergic neuronal NMDARs can rebalance hippocampal networks and highlights EU1622-240 as a promising therapeutic avenue for disorders characterized by NMDAR hypofunction.

However, I have a few concerns that should be addressed before publication.

Majors :

1. Figure 6 & Page 16 line 20: Given that there are other subtypes of NMDARs in CA1 interneurons and EU1622-240 can potentiate both GluN2B and GluN2D, there is probably synergism of GluN2B and GluN2D in improving interneuron excitability. To dissect the relative contribution of each subunit, experiments in GluN2B-KO (or GluN2B/GluN2D double-KO)

interneurons, together with quantitative immunostaining for GluN2B versus GluN2D, are needed. These additional controls will clarify whether the GluN2D-dependence observed in Figure 6 fully accounts for the drug effect or whether GluN2B also participates.

2. Figure 7: antagonists of NMDARs or GluN2D -KO hippocampal neurons should be applied to confirm that the effects of EU1622-240 on Ca²⁺ entry act via NMDARs.

3. Page 19 line 2: The conclusion that "3 μ M 1622-240 disrupted potentiation of the slope of fEPSP in response to theta burst stimulation" should be further studied because it's not significant ($p=0.0992>0.05$). The authors should either increase the sample size, apply a more sensitive statistical test, or revise the wording to reflect the lack of statistical support.

4. The study pools all CA1 stratum radiatum (s.r.) interneurons as a single population. However, the hippocampus contains heterogeneous GABAergic subtypes (PV+, CCK+, SOM+, VIP+ cells) that differ in morphology, intrinsic physiology, synaptic wiring, and GluN2 subunit expression (PMID: 37467748). These GABAergic interneurons all influence the excitability of CA1 pyramidal neurons. More evidence is needed to determine whether EU1622-240 preferentially modulates particular interneuron subclasses.

Minors :

1. Figure 2 D1/ Table 1: Page 14 line 26 says that "Interneurons showed no detectable increase in the mean weighted time constant (τ_w) for deactivation". However, Table 1 shows that the p-value of τ_w of interneuron is $0.0200<0.05$, the panel should be labelled with "***".

2. Figure 7: The plotting style should be consistent with other figures. For example, the font used for labels is not bold.

3. Figure 10: the font of " μ " in this figure (B-D & F-G) is inconsistent with that in other figures. There are no figure legends of figure 10. D-H.

4. The Results text would benefit from careful language polishing to improve clarity and flow.

Referee #2:

NMDA receptors play critical roles in brain excitation and dysfunction and identifying means to modulate this activity is critical for potential therapeutic treatments. In this work, the authors explore the mechanism of action of EU1622-240, a positive allosteric modulator of GluN2-containing NMDARs, mainly GluN2B, GluN2C and GluN2D, in the context of the hippocampal circuit. This study is of great clinical importance as the hypofunction of NMDARs is associated with multiple neurological disorders and defining how compounds work in a circuit is critical to defining their potential therapeutic value. Here, the authors focus on the effect of this compound on the excitability of both excitatory cells and interneurons in the hippocampus. They uncover that EU1622-240 primarily enhances the activity of interneurons via acting on the GluN2D-containing NMDA receptors. Actions on excitatory pyramidal neurons is less notable due to enhanced GABA input.

The manuscript is presented in a very clear and readable form. The analysis is rigorous and detailed. The results are clearly presented as are the figures. Statistical analysis is highly appropriate. We only have some minor suggestions for improvement/clarity.

Comments.

1. The described action of EU1622-240 specifically in the hippocampal interneurons seems to be primarily specific to its action on GluN2D-containing receptors. Perhaps, this point should be noted earlier in the manuscript for example in the Abstract. It feels like it just appears late in the manuscript. In fact when I was first looking at early figures (Figures 2-5), I felt like they should directly test the role of GluN2D in the action EU1622-240 on interneurons, only to discover later (Figure 6) that this was actually tested.

2. The reason for the focus on immature hippocampus, as opposed to mature, and interneurons specifically in the CA1 stratum radiatum is not specified. The early time point is relevant to neurodevelopmental disorders - schizophrenia and epilepsy as mentioned. But there is also an emphasis on learning, memory, and Alzheimer's disease, which applies to a

wide age-range.

3. Are the Ca²⁺ imaging experiments specific to PCs? Moreover, was there a reason not to conduct them in the GluN2D-KO mice to parse apart the contribution of GluN2D vs GluN2B to EU1622-240 mechanism of action?

4. Pg. 17: "Taken together, these data show that there are increases of both inhibitory and excitatory signal tone onto CA1 pyramidal cells during EU1622-240 application." How does the Ca²⁺ imaging data support this claim is all that is being show is that EU1622-240 increased calcium flux into dissociated neurons?

Minor points

1. Pg. 7: The authors write: "The NMDAR is a heterotetrametric complex of two obligate GluN1 subunits and two GluN2 subunits." The existence of GluN3 subunit should probably be mentioned as well.

2. Pg. 9: "The brain was then hemisected, glued to the removable stage and sectioned on a vibratome into (Leica VT1200S, Wetzlar, Germany)." Sentence appears to get cut off.

3. Pg. 12: "Microscopy and electrophysiology experiments were performed within 24 hr of transfection." Was electrophysiology conducted on the cultures?

4. Figure 1B, right panel: Cell bodies are extremely hard to see. Perhaps outline?

5. Figure 2: Perhaps change the scale of panel B to match that of A to further highlight that the effect is greater on INs than PCs. Perhaps switch panel E and D as E is referenced in the text prior to D.

6. Pg. 14: "For both interneurons and CA1 pyramidal cells, we observed a modest increase in EPSC amplitude and charge transfer (Fig. 2, Table 1)." Should indicate Fig. 2C, E, Table 1.

7. Figure 3: Is the IN increase in firing frequency and RMP depolarization significant across concentrations of EU240?

8. Figure 4C: Significant of 6uM EU1622-240 relative to 0 uM is not highlighted.

9. Figure 3C: The pyramidal cell RMP seems to trend downwards with increasing EU240 concentrations. I assume this result is not significant? What could account for it? Increased inhibitory drive onto PCs in the presence of this compound, leading to a slight decrease in PC RMP?

10. Pg. 15: "In the presence of EU1622-240, injection of hyperpolarizing current failed to reduce firing (Fig. 5A2)," Suppress may be a better word than reduce.

11. Figure 5: What is the average rheobase current in all the experimental conditions?

12. Pg. 16: "As expected, we observed a robust increase in QCT amplitude following application of 3 μ M EU1622-240, with a similar increase in the integrated area under the Ca²⁺ imaging traces, which reflects QCT amplitude, frequency, and duration over the imaging session (Fig. 7E)." Reference panels C & D as well?

13. Pg. 17: "We detected both depolarization of RMP and increased spontaneous firing in CA1 pyramidal cells with co-administration of bicuculline and EU1622-240 (Fig. 8A-C)." Reference panel D as well?

14. Pg. 17: "One potential interpretation is that the increased tonic inhibition produced by enhanced NMDAR drive onto interneurons masks a depolarizing effect of EU1622-240 on CA1 pyramidal cells." Could be worthwhile to mention that the depolarizing effect of EU1622-240 on PCs is, thus, likely due to GluN2B-containing NMDARs.

15. Pg. 17: EU1622-240 significantly increased IPSC frequency to 1.8 {plus minus} 0.56-fold of control (n = 8; Fig. 9B,C unpaired t-test, p = 0.000169 < 0.025). By contrast, there was no effect of EU1622-240 on IPSC frequency when APV was present to block NMDARs (Treatment/Control was 0.76 {plus minus} 0.48-fold, n = 14, p = 0.34). There was no detectable difference in sIPSC amplitude in EU1622-240 in the absence (0.91 {plus minus} 0.21-fold of baseline, paired t-test, p = 0.086) or in the presence of APV (1.0 {plus minus} 0.22-fold of baseline, p = 0.33). When parallel experiments were performed with 400 μ M D,L-APV present in the aCSF, APV itself did not detectably alter mean sIPSC frequency (control 8.3 {plus minus} 4.1 Hz, APV 9.0 {plus minus} 6.0 Hz, paired t-test, p = 0.307, Fig. 9B)." Are these experiments all being done with D,L-APV? It is unclear.

16. Figure 10: Label panel A as distal stimulation and panel E as local stimulation? In addition the description of of panels E-H is missing.

17. Pg. 18: "3 μ M EU1622- 240 significantly increased the fold change of field amplitude (repeated-measure one-way ANOVA with Dunnett multiple comparison test: F (1.178, 8.247) = 6.653, p = 0.0285; Fig. 11C)." Significance not shown on figure.

18. Figure 11E: Is the result across the two groups significant?

19. Pg. 19: "The preferential enhancement of interneuron excitability by EU1622-240 could reflect the higher firing rate of interneurons, which could reduce Mg²⁺ blockade of NMDARs, allowing for a higher level of basal NMDAR synaptic currents on interneurons (Cohen, Tsien et al. 2015)." I believe the point being made here is that the higher firing rate of neurons reflects a more depolarized resting membrane potential, leading to decreased Mg²⁺ block. As written, it reads like the increased firing rate, which would leads to increased inhibition, decreases Mg²⁺ block.

END OF COMMENTS

Responses to Reviewers – Manuscript ID: JP-RP-2025-289774

We thank the handling editor and the two expert referees for their careful and thoughtful assessment of our manuscript. Their considered feedback and valuable recommendations have strengthened both the substance and presentation of this study.

Reviewing Editor Comments:

Overall both reviewers consider that this manuscript is of high impact to the field and have provided comments for authors to revise the manuscript. Please carefully read the reviews, revise the manuscript (including additional experiments if necessary) and submit a revised manuscript together with a point-to-point response letter. Please also see 'Required Items' above.

Thank you for taking the time to review our manuscript. We have provided a point-by-point response to the reviewers' comments. We have reviewed the checklist and confirmed that we are fully compliant with Journal policy.

Senior Editor Comments:

Thank you for submitting your manuscript to The Journal of Physiology. Your study on the selective modulation of GluN2D-containing NMDARs by EU1622-240 in hippocampal microcircuits has been evaluated by two expert reviewers and a reviewing editor. Both reviewers and the editor found your work to be of high quality and potential interest to the field, but they have raised several substantive concerns that need to be addressed before the manuscript can be considered for acceptance. Please submit a revised manuscript with a point-by-point response to the reviewers' comments. Please consider clarifying the relative contributions of GluN2B and GluN2D to EU1622-240's effects (on Ca²⁺ entry) by including experiments in GluN2B-KO mice or immunostaining for these subunits.

The goal of this study was not to establish subunit-selectivity for this series, but to evaluate the actions of this pan-PAM at native synapses in the context of microcircuitry. The calcium imaging experiment (Ca²⁺ entry assay) was designed to investigate whether EU1622-240 application potentiates excitatory neurotransmission onto pyramidal neurons via GluN2B-containing NMDA receptors. Cultured pyramidal cells are known to express GluN2B, with little or no GluN2D expression. Repeating the Ca²⁺ entry experiment in cultured pyramidal cells obtained from GluN2B-KO mice would provide a negative control, but these mice are not viable (they die unless manually fed as neonates), making maintenance of a colony challenging. There are floxed 2B, but our previous experience with this colony was that they were hard to maintain, and we would have to re-establish this colony (obtaining mice from UC Davis) and cross it with the appropriate CRE driver to do these experiments, which would require many months. It is also unknown whether global knockout of GluN2B alters NMDA receptor subunit composition in pyramidal cells (we have shown that global knockout of GluN2A promotes compensation of mRNA encoding other

NMDA receptor subunits, Camp et. al. 2023). We could attempt to knockdown GluN2B in cultured neurons via shRNA, however, these experiments usually do not result in 100% knockdown, making conclusions drawn from these experiments less secure. Since we are not attempting to describe this compound as subunit-specific, we have chosen to show application of 1622-240 can indeed potentiate a variety of NMDA receptor-mediated responses in interneurons and in principal cells, as a first step to understanding the actions of this class of modulator in the context of native receptors and circuits.

Referee #1 Comments:

Xing and colleagues' research significantly advances our understanding of NMDA receptor (NMDAR) modulation in hippocampal microcircuits. Their study demonstrates that the positive allosteric modulator EU1622-240 selectively enhances the excitability of CA1 stratum radiatum interneurons while leaving CA1 pyramidal cells largely unaffected. Using whole-cell patch-clamp recordings, calcium imaging, and field-potential assays in mouse hippocampal slices and cultured neurons, they show that EU1622-240 potentiates GluN2D-containing NMDARs expressed predominantly on interneurons, leading to membrane depolarization, increased spike firing, and elevated inhibitory output onto pyramidal cells. This shift in the excitation-inhibition balance is accompanied by a marked rise in spontaneous inhibitory postsynaptic currents (sIPSCs) recorded from CA1 pyramidal cells. In addition, the compound induces a long-lasting potentiation of synaptic transmission that occludes conventional theta-burst LTP, suggesting a novel form of NMDAR-dependent plasticity. Critically, these effects are abolished in GluN2D-knockout mice, confirming the subunit specificity of EU1622-240. Overall, this work provides compelling evidence that selective potentiation of GABAergic neuronal NMDARs can rebalance hippocampal networks and highlights EU1622-240 as a promising therapeutic avenue for disorders characterized by NMDAR hypofunction. However, I have a few concerns that should be addressed before publication.

Major:

1. Figure 6 & Page 16 line 20: Given that there are other subtypes of NMDARs in CA1 interneurons and EU1622-240 can potentiate both GluN2B and GluN2D, there is probably synergism of GluN2B and GluN2D in improving interneuron excitability. To dissect the relative contribution of each subunit, experiments in GluN2B-KO (or GluN2B/GluN2D double-KO) interneurons, together with quantitative immunostaining for GluN2B versus GluN2D, are needed. These additional controls will clarify whether the GluN2D-dependence observed in Figure 6 fully accounts for the drug effect or whether GluN2B also participates.

The reviewer is correct in their understanding that EU1622-240 potentiates GluN2B, which is confirmed in Fig. 2 B, C2, & E2, given that GluN2B is on CA1 pyramidal cells. While the relative contribution of GluN2B's actions to the interneuron's response to EU1622-240 have not been assessed, the suggested experiments are simply beyond the scope of the manuscript, which already has 11 figures, multiple tables, and has taken ~5 years to complete. The GluN2B KO mice are not viable, and thus we would need to obtain the conditional floxed-GluN2B mice via MTA from UC-Davis, and then cross these with an

appropriate driver line, and then cross the resulting mice with GluN2D KO mice (three crosses, taking nearly a year). We have neither the personnel nor funding to undertake these experiments, which would only marginally advance the conclusions. We predict we would see modestly diminished effects of 1622-240 in GluN2B KO mice because GluN2A could take over the place of GluN2B in triheteromeric receptor. Furthermore, the potentiation via GluN2B has not been robust enough to generate measurable changes in CA1 pyramidal cells firing under physiological condition (Fig. 3 C, Fig. 4 D-F), suggesting a modest role in the absence of GluN2D. That is, EU1622-240 failed to depolarize interneurons from GluN2D KO mice, even though GluN2B is present or potentially upregulated (PMID: 37031803). We believe the current data provides adequate support for our conclusion that EU1622-240's enhancement of interneuron function is dependent on GluN2D expression.

2. Figure 7: antagonists of NMDARs or GluN2D KO hippocampal neurons should be applied to confirm that the effects of EU1622-240 on Ca²⁺ entry act via NMDARs.

We agree with this suggestion, and have conducted an additional experiment where we show that APV/MK801 treatment completely blocks Ca²⁺ entry under these conditions, even following potentiation by EU1622-240. These data are now included in Fig. 7 of the revised manuscript. We also note that we performed our experiments in the presence of nimodipine to eliminate the potential involvement of L-type VGCCs, a major source of dendritic Ca²⁺ spikes.

3. Page 19 line 2: The conclusion that "3 μM 1622-240 disrupted potentiation of the slope of fEPSP in response to theta burst stimulation" should be further studied because it's not significant ($p=0.0992>0.05$). The authors should either increase the sample size, apply a more sensitive statistical test, or revise the wording to reflect the lack of statistical support.

I believe the result is correct as stated, as the drug disrupted LTP (i.e. there was no significant LTP, $p<0.0992$). So, the lack of significance supports the drug-induced disruption (elimination) of LTP.

4. The study pools all CA1 stratum radiatum (s.r.) interneurons as a single population. However, the hippocampus contains heterogeneous GABAergic subtypes (PV+, CCK+, SOM+, VIP+ cells) that differ in morphology, intrinsic physiology, synaptic wiring, and GluN2 subunit expression (PMID: 37467748). These GABAergic interneurons all influence the excitability of CA1 pyramidal neurons. More evidence is needed to determine whether EU1622-240 preferentially modulates particular interneuron subclasses.

We fully agree with the referee's point that multiple subtypes of CA1 interneurons exist with different levels/combinations of NMDA receptor subunit expression. Considering that CA1 interneurons generally express GluN2D (PMID: 27625038), this manuscript did not investigate the EU1622-240 effect onto specific interneuron subtypes. Rather, this question is a topic for future work, and we have applied for funding to do this. However, to obtain information on identified interneuron subtypes in this manuscript (which already has 11 figures) would require us to repeat recordings performed in slices 4 or more times (PV interneurons, VIP interneurons, SST interneurons, CCK interneurons, NPY neurogliaform cells, etc). If the reviewer then wanted an assessment of, say GluN2B using a selective inhibitor, we would need even more recordings for each cell type. Instead, our goal was to obtain a broad view of EU1622-240

modulator's effects on CA1 interneurons to provide the field and ourselves a starting point from which to focus on specific interneuron subtypes in the context of this unique class of modulator. Thus, we consider this approach, while valuable, simply beyond the scope of this manuscript, which already took considerable time and effort to complete and includes a large amount of data.

Minor

1. Figure 2 D1/ Table 1: Page 14 line 26 says that "Interneurons showed no detectable increase in the mean weighted time constant (Tauw) for deactivation". However, Table 1 shows that the p-value of Tauw of interneuron is $0.0200 < 0.05$, the panel should be labelled with "*".

The analysis of response time course in Fig 2/Table 1 includes assessment of multiple parameters (EPSC amplitude, Tauw, and charge transfer) from the same dataset (evoked-EPSCs from CA1 interneurons). Thus, we must apply a family-wise error rate (FWER) correction. We used the Holm-Bonferroni FWER method, and thus the p-value for Tauw was ranked as the 3rd lowest, which needs to be smaller than 0.0125 (0.05 divided by 2, and divided by 2 again for 3rd lowest p-value) to be considered significant. Our p-value here is $0.02 > 0.0125$. It is thus not significant. The figure and Table are correct. We have expanded our description of the family-wise error in our methods.

2. Figure 7: The plotting style should be consistent with other figures. For example, the font used for labels is not bold.

We have modified the figure as requested.

3. Figure 10: the font of " μ " in this figure (B-D & F-G) is inconsistent with that in other figures. There are no figure legends of figure 10. D-H.

Thanks for pointing out this typo. The figure label font has been corrected to be consistent, and figure legend for figure 10. D-H has been updated.

4. The Results text would benefit from careful language polishing to improve clarity and flow.

We thank the referee for their careful review and suggestions. The results section has now been edited and proofread to give better flow and clarity for readers.

Referee #2 Comments:

NMDA receptors play critical roles in brain excitation and dysfunction and identifying means to modulate this activity is critical for potential therapeutic treatments. In this work, the authors explore the mechanism of action of EU1622-240, a positive allosteric modulator of GluN2-containing NMDARs, mainly GluN2B, GluN2C and GluN2D, in the context of the hippocampal circuit. This study is of great clinical importance as the hypofunction of NMDARs is associated with multiple neurological disorders and defining how compounds work in a circuit is critical to defining their potential therapeutic value.

Here, the authors focus on the effect of this compound on the excitability of both excitatory cells and interneurons in the hippocampus. They uncover that EU1622-240 primarily enhances the activity of interneurons via acting on the GluN2D-containing NMDA receptors. Actions on excitatory pyramidal neurons is less notable due to enhanced GABA input.

The manuscript is presented in a very clear and readable form. The analysis is rigorous and detailed. The results are clearly presented as are the figures. Statistical analysis is highly appropriate. We only have some minor suggestions for improvement/clarity.

1. The described action of EU1622-240 specifically in the hippocampal interneurons seems to be primarily specific to its action on GluN2D-containing receptors. Perhaps, this point should be noted earlier in the manuscript for example in the Abstract. It feels like it just appears late in the manuscript. In fact when I was first looking at early figures (Figures 2-5), I felt like they should directly test the role of GluN2D in the action EU1622-240 on interneurons, only to discover later (Figure 6) that this was actually tested.

Thank you for the thoughtful suggestion regarding the data presentation flow of this manuscript. We have updated the Abstract by pointing out the role of GluN2D-containing NMDA receptors during our EU1622-240 effects so that it is clear from the outset that we will assess interneuron function.

2. The reason for the focus on immature hippocampus, as opposed to mature, and interneurons specifically in the CA1 stratum radiatum is not specified. The early time point is relevant to neurodevelopmental disorders - schizophrenia and epilepsy as mentioned. But there is also an emphasis on learning, memory, and Alzheimer's disease, which applies to a wide age-range.

Thank you for pointing this out. We have updated the manuscript with new text at the end of the discussion to explain the rationale for our focus on the P17-P22 age range hippocampal neurons regarding the early and late time point related disease. We now raise the idea that additional work will be needed to assess the response of circuits in older animals to this class of modulator.

3. Are the Ca²⁺ imaging experiments specific to PCs? Moreover, was there a reason not to conduct them in the GluN2D-KO mice to parse apart the contribution of GluN2D vs GluN2B to EU1622-240 mechanism of action?

Yes, the Ca²⁺ imaging experiment was specific to pyramidal cells. The purpose of Ca²⁺ imaging experiment here is to test if there is excitatory tone onto pyramidal cells via direct GluN2B potentiation during EU1622-240 treatment. Wild type CA1 pyramidal cells express abundant GluN2B (which will be potentiated by our EU1622-240) and not GluN2D. Cultured pyramidal cells from GluN2D-KO mice will almost certainly exhibit similar GluN2A/B subunit expression profiles (i.e. abundant GluN2B), and thus it is difficult to see how much will be gained from repeating these experiments in cultures made from GluN2D KO mice. For this reason, we did not include this permutation of our experimental design.

4. Pg. 17: "Taken together, these data show that there are increases of both inhibitory and excitatory signal tone onto CA1 pyramidal cells during EU1622-240 application." How does the Ca²⁺ imaging data support this claim is all that is being show is that EU1622-240 increased calcium flux into dissociated neurons?

We originally intended to convey the ideas that EU1622-240 could enhance excitatory drive onto these cultured neurons. However, we appreciate the referee's perspective here and have removed this comment.

Minor points

1. Pg. 7: The authors write: "The NMDAR is a heterotetrametric complex of two obligate GluN1 subunits and two GluN2 subunits." The existence of GluN3 subunit should probably be mentioned as well.

The sentence has been updated to include GluN3 subunit in the manuscript.

2. Pg. 9: "The brain was then hemisected, glued to the removable stage and sectioned on a vibratome into (Leica VT1200S, Wetzlar, Germany)." Sentence appears to get cut off.

Thanks for pointing out this typo. The sentence has been corrected.

3. Pg. 12: "Microscopy and electrophysiology experiments were performed within 24 hr of transfection." Was electrophysiology conducted on the cultures?

There was no electrophysiology conducted on cultured neurons. The sentence has been corrected.

4. Figure 1B, right panel: Cell bodies are extremely hard to see. Perhaps outline?

Thanks for the suggestion. We have added outlines for pyramidal cells (red outline) and interneuron (blue outline) and updated the Fig.1 legend in the manuscript to mention this.

5. Figure 2: Perhaps change the scale of panel B to match that of A to further highlight that the effect is greater on INs than PCs. Perhaps switch panel E and D as E is referenced in the text prior to D.

We appreciate the referee's suggestions. We chose to use different scales between panel A and panel B, to maximize the size of the current traces so the reader could fully appreciate the response waveform and have the clearest view of the evoked-EPSC current changes before (black) and after (blue) EU1622-240 treatment on CA1 interneuron and pyramidal cell, respectively. We present Fig. 2D ahead of Fig. 2E because Fig.2E was calculated from Fig.2C & 2D, and it is reasonable to present EPSC amplitude and Tauw, then subsequently present data (charge transfer) calculated from amplitude and Tauw. In the manuscript text, we have modified how we refer to the figures so that it does not appear any panel is presented ahead of another in error.

6. Pg. 14: "For both interneurons and CA1 pyramidal cells, we observed a modest increase in EPSC amplitude and charge transfer (Fig. 2, Table 1)." Should indicate Fig. 2C, E, Table 1.

Thanks for the correction. The manuscript has been updated accordingly.

7. Figure 3: Is the IN increase in firing frequency and RMP depolarization significant across concentrations of EU240?

Yes, as we show in Fig. 4A, interneurons increase their firing frequency during 3 and 6 μ M EU1622-240 administration but not at lower concentrations. RMP was significantly depolarized at 6 μ M, but not at lower concentrations.

8. Figure 4C: Significant of 6 μ M EU1622-240 relative to 0 μ M is not highlighted.

We did not highlight significance of 6 μ M to 0 μ M because it is not significant. We chose to only show whenever there is significance to avoid the figures being overcrowded by “ns” labels, which potentially distract readers from observing the significant effects.

9. Figure 3C: The pyramidal cell RMP seems to trend downwards with increasing EU240 concentrations. I assume this result is not significant? What could account for it? Increased inhibitory drive onto PCs in the presence of this compound, leading to a slight decrease in PC RMP?

Thanks for this question. 1) Correct, the RMP for pyramidal cells is not significantly changed by EU1622-240. 2) We agree with the referee here, and we confirm that there is an increased inhibitory drive onto pyramidal cells in the presence of EU1622-240 (Fig. 8 and Fig. 9), which we hypothesize overwhelms the GluN2B-mediated depolarization effect of EU1622-240 on pyramidal cells. We have tried to make this point more clearly in the revised text.

10. Pg. 15: "In the presence of EU1622-240, injection of hyperpolarizing current failed to reduce firing (Fig. 5A2)," Suppress may be a better word than reduce.

Thanks for the suggested improvement in wording. The word “suppress” has been used in this sentence.

11. Figure 5: What is the average rheobase current in all the experimental conditions?

Unfortunately, the smallest step current injection (+12 pA) we used in this experiment in Fig. 5 was too large for a comprehensive rheobase assessment (current injections steps should be \sim 2 pA). Thus, we did not obtain information on rheobase in our experimental conditions.

12. Pg. 16: "As expected, we observed a robust increase in QCT amplitude following application of 3 μ M EU1622-240, with a similar increase in the integrated area under the Ca²⁺ imaging traces, which reflects QCT amplitude, frequency, and duration over the imaging session (Fig. 7E)." Reference panels C & D as well?

Done. Corrected references have been made in manuscript.

13. Pg. 17: "We detected both depolarization of RMP and increased spontaneous firing in CA1 pyramidal cells with co-administration of bicuculline and EU1622-240 (Fig. 8A-C)." Reference panel D as well?

Yes, we agree with referee’s suggestion. The correction has been made in the manuscript.

14. Pg. 17: "One potential interpretation is that the increased tonic inhibition produced by enhanced NMDAR drive onto interneurons masks a depolarizing effect of EU1622-240 on CA1 pyramidal cells." Could be worthwhile to mention that the depolarizing effect of EU1622-240 on PCs is, thus, likely due to GluN2B-containing NMDARs.

Yes, it is worth mentioning this. We have updated the manuscript accordingly.

15. Pg. 17: EU1622-240 significantly increased IPSC frequency to 1.8 {plus minus} 0.56-fold of control (n = 8; Fig. 9B,C unpaired t-test, p = 0.000169 < 0.025). By contrast, there was no effect of EU1622-240 on IPSC frequency when APV was present to block NMDARs (Treatment/Control was 0.76 {plus minus} 0.48-fold, n = 14, p = 0.34). There was no detectable difference in sIPSC amplitude in EU1622-240 in the absence (0.91 {plus minus} 0.21-fold of baseline, paired t-test, p = 0.086) or in the presence of APV (1.0 {plus minus} 0.22-fold of baseline, p = 0.33). When parallel experiments were performed with 400 μ M D,L-APV present in the aCSF, APV itself did not detectably alter mean sIPSC frequency (control 8.3 {plus minus} 4.1 Hz, APV 9.0 {plus minus} 6.0 Hz, paired t-test, p = 0.307, Fig. 9B)." Are these experiments all being done with D,L-APV? It is unclear.

Yes, D,L-APV was the only "APV" used in our experiments. We have modified the text to more clearly indicate D,L-APV was used. Thanks for pointing ambiguity this out.

16. Figure 10: Label panel A as distal stimulation and panel E as local stimulation? In addition the description of of panels E-H is missing.

Yes, panel A is distal stimulation and panel E is local stimulation. We have updated the Fig. 10 legend to clarify this.

17. Pg. 18: "3 μ M EU1622- 240 significantly increased the fold change of field amplitude (repeated-measure one-way ANOVA with Dunnett multiple comparison test: F (1.178, 8.247) = 6.653, p = 0.0285; Fig. 11C)." Significance not shown on figure.

Thanks for spotting this. A significance asterisk has been added onto figure 11C.

18. Figure 11E: Is the result across the two groups significant?

No, the result across the two groups was not significant, as the lowest p-value is 0.0992. This is consistent with our conclusion that there is no detectable LTP after addition of EU1622-240.

19. Pg. 19: "The preferential enhancement of interneuron excitability by EU1622-240 could reflect the higher firing rate of interneurons, which could reduce Mg²⁺ blockade of NMDARs, allowing for a higher level of basal NMDAR synaptic currents on interneurons (Cohen, Tsien et al. 2015)." I believe the point being made here is that the higher firing rate of neurons reflects a more depolarized resting membrane potential, leading to decreased Mg²⁺ block. As written, it reads like the increased firing rate, which would leads to increased inhibition, decreases Mg²⁺ block.

Yes, the referee's interpretation is correct. We have modified the sentence to avoid any confusion as "The preferential enhancement of interneuron excitability by EU1622-240 could reflect the higher firing rate of interneurons with a more depolarized resting membrane potential, which could reduce Mg²⁺ blockade of NMDARs located on these interneurons, allowing for a higher level of basal NMDAR synaptic currents onto interneurons (Cohen, Tsien et al. 2015)." This has been updated in the manuscript.

Dear Dr Traynelis,

Re: JP-RP-2025-289774R1 "**Enhancement of hippocampal interneuron excitability by NMDA receptor positive allosteric modulation**" by Hao Xing, Tue G. Banke, Lu Zhang, Kuai Yu, Chad R. Camp, Russell G Fritzsche, Nicholas Akins, Srinu Paladugu, Paul J Arcoria, Brian R Brady, Olga Prikhodko, Matthew Kennedy, Dennis C. Liotta, HONGJIE YUAN, and Stephen F. Traynelis

We are pleased to tell you that your paper has been accepted for publication in The Journal of Physiology.

Yours sincerely,

Jing-Ning Zhu
Senior Editor
The Journal of Physiology

If you would like to receive our 'Research Roundup', a monthly newsletter highlighting the cutting-edge research published in The Physiological Society's family of journals (The Journal of Physiology, Experimental Physiology, Physiological Reports, The Journal of Nutritional Physiology and The Journal of Precision Medicine: Health and Disease), please click this link, fill in your name and email address and select 'Research Roundup':
<https://www.physoc.org/journals-and-media/membernews>

- You can help your research get the attention it deserves! Check out Wiley's free Promotion Guide for best-practice recommendations for promoting your work at: www.wileyauthors.com/eoo/guide. You can learn more about Wiley Editing Services which offers professional video, design, and writing services to create shareable video abstracts, infographics, conference posters, lay summaries, and research news stories for your research at: www.wileyauthors.com/eoo/promotion.

EDITOR COMMENTS

Reviewing Editor:

I am delighted to see that the referees are satisfied with your responses and I am happy to recommend acceptance of your manuscript for publication.

Senior Editor:

The authors address all concerns raised by the reviewers and editors.

REFEREE COMMENTS

Referee #1:

I have no more questions and agree to accept this article.

Referee #2:

The authors have done an extremely nice job of addressing the Editor's and the Reviewers' comments. The experiments were rigorous, interesting and important and are described in a clear fashion. The rigorous accommodation of the various comments further enhances the quality of an extremely nice paper. I have no further comments for the authors.